# A benchmark of semi-supervised scRNA-seq integration methods in real-world scenarios

Xiaoyu Shen[1], Chuan He[2]*, Leying Guan[2,3]*

**1** Department of Statistics and Data Science, Yale University, New Haven, Connecticut, United States of America, **2** Department of Biostatistics, Yale School of Public Health, New Haven, Connecticut, United States of America, **3** Program in Computational Biology and Bioinformatics, Yale University, New Haven, Connecticut, United States of America

\* ch2343@yale.edu (CH); leying.guan@yale.edu (LG)

## Abstract

Semi-supervised methods for single-cell RNA-seq integration promise improved batch correction and preservation of biological signal by leveraging cell-type labels. However, reported benefits and robustness of them towards imperfect cell type labels often come from overly idealized settings. Here we present, to our knowledge, the first systematic benchmark comparing leading semi-supervised methods with widely used unsupervised approaches across six diverse datasets under realistic conditions. Beyond randomly missing or erroneous labels, we examine four additional scenarios (boundary-mixed labels, batch-specific annotations, auto-generated labels and varied-granularity labels) and evaluate performance using nine established metrics. We find that although semi-supervised methods can provide benefits under perfect annotations, their robustness often degrades substantially under realistic imperfections. Only scANVI and ssSTACAS maintain stable but modest improvements over their unsupervised counterparts, and none consistently outperform the strongest unsupervised approach. These results indicate that current semi-supervised strategies offer limited practical advantage when label quality is modest uncertain.

### Author summary

Single-cell RNA sequencing (scRNA-seq) offers deep insights into cellular diversity, yet combining datasets from different sources remains difficult due to technical "batch effects." While semi-supervised integration methods promise to improve alignment by utilizing cell-type labels, their reliability under realistic, imperfect conditions remains unproven. We performed a comprehensive benchmark comparing leading semi-supervised algorithms against standard unsupervised approaches using realistic scenarios, such as randomly missing, mixing at edge, partially annotated by batches, automated labeling and varied-granularity labels. Our results overturn the assumption that adding labels always improves

**Data availability statement:** Data Availability: All data are within the manuscript and supporting information. Code Availability: https://github.com/RainySheena/benchmark_semi.

**Funding:** This work was supported by the National Science Foundation (DMS2310836 to L.G.) and the National Institutes of Health (5U01AI167892 to L.G.). The funders had no role in study design, data collection and analysis, decision to publish, or preparation of the manuscript.

**Competing interests:** The authors have declared that no competing interests exist.

integration. We found that methods relying heavily on labels, such as scDREAM-ER, excel with perfect or Azimuth-generated annotations but degrade rapidly with structured errors. Conversely, robust supervision methods like scANVI and ssSTACAS are stable but offer limited advantages. Based on these results, we recommend scDREAMER when annotations are high confidence, a state-of-the-art unsupervised method such as scCRAFT as the most reliable default when labels are incomplete or noisy, and scANVI as a robust semi-supervised alternative when partial labels are trusted.

## 1 Introduction

Single-cell RNA sequencing (scRNA-seq) has revolutionized our understanding of cellular diversity and resulted in large-scale resources that capture cellular states under various biological conditions and individuals [1–3]. Initiatives such as the Human Cell Atlas [4] and the Human BioMolecular Atlas Program [5], which aim to map human tissues at unprecedented resolution, have provided valuable insights into cellular heterogeneity in health and disease. However, these large-scale scRNA-seq resources are often affected by batch effects–unwanted technical and biological variation introduced by differences in sample handling, sequencing platforms, protocols, or inter-individual factors such as microenvironmental context [6–9]. To address this challenge, a wide range of computational integration methods have been developed to mitigate batch-specific transcriptomic shifts. Most integration methods are unsupervised and rely solely on the count matrices of the scRNA-seq data and their performance has been extensively benchmarked [10].

Semi-supervised integration methods leverage available cell-type labels to improve embedding quality and batch correction by guiding the alignment process with partial supervision. For example, scANVI [11] and scDREAMER [12] encourage accurate classification of annotated cells to their labels during integration. scGEN [13] models gene expression changes associated with discrete covariates such as batch by learning latent shifts across conditions, which are estimated from matched or re-sampled cell-type distributions to isolate the effect of interest. ItClust [14] applies transfer learning to propagate information from labeled reference cells to unlabeled target cells. More recently, ssSTACAS [15] refines the mutual nearest neighbor anchoring process by pruning cell pairs whose labels are incompatible, thereby improving alignment fidelity. Although improvement of semi-supervised approaches have been observed in their respective studies, their evaluations often rely on idealized scenarios with complete or randomly perturbed labels, failing to reflect real-world challenges where annotations are incomplete, inconsistent, erroneous, or automated with variable accuracy [16–18]. Consequently, the true utility and robustness of these semi-supervised methods in practical applications remain poorly understood.

To bridge this gap, we present a comprehensive benchmark of semi-supervised single-cell RNA-sequencing integration methods under realistic conditions. We systematically evaluate five leading semi-supervised methods (scANVI, scGEN,

ssSTACAS, scDREAMER, and ItClust) against five unsupervised methods (Seurat RPCA [19], scVI [20], Harmony [21], Scanorama [22], and scCRAFT [23]) that are selected based on popularity and reported performance (see Section 1 in S1 Text for detailed description of each method). Our benchmark design incorporates two commonly examined conditions: (1) randomly missing labels and (2) randomly wrong labels, as well as four realistic scenarios that mimic practical challenges: (3) ambiguous labels at cluster boundaries, (4) batch-specific partial annotations, (5) labels generated by automated annotation tools and (6) labels with varied granularity.

By assessing semi-supervised and unsupervised methods in different annotation settings using six diverse datasets spanning multiple tissues, organs, and species, our study provides the first systematic and practically grounded evaluation of semi-supervised integration strategies, offering critical guidance for their effective use in real-world single-cell analyses.

## 2 Results

### 2.1 Benchmark design overview

This study considers in total six annotation scenarios. The first is a baseline setting where unsupervised methods perform integration, and supervised methods are provided with correct labels for all cells. The other five are partial label scenarios, two of which have been widely used in previous work to test the robustness of semi-supervised methods: one with randomly missing labels and another with randomly incorrect labels, each at varying degrees. (Fig 1).

The third partial label scenario is *missing and mixing at edge*. A key factor in achieving strong biological conservation is the separation of such closely related cell types where even a small number of correctly annotated cells can help reinforce this distinction and improve integration performance [10]. However, misannotations in practice often arise specifically at these boundaries [23]. To account for this, we introduces the scenario where labels are mixed between transcriptionally similar cell types, rather than randomizing labels arbitrarily. Concretely, in the *missing and mixing at edge* scenario, we first conduct unsupervised integration with Harmony, then, we examine the 30 nearest neighbors for each cell and compute the proportion of neighbors that share its true cell type. If this proportion falls below a threshold $\gamma \in [0, 1]$, then the cell's label is reassigned to a different type sampled from its neighbors' label distribution. If the resampled label still matches the cell's true type, we instead mark the label as "unknown." Thus, a higher $\gamma$ value selects more cells for label alteration. Each selected cell's label is then changed to a neighboring cell type or "unknown." Consequently, increasing $\gamma$ results in more significant label corruption and a more difficult annotation task.

The fourth partial label scenario, *partially annotated batches*, simulates situations where only a randomly selected subset of batches is labeled. This reflects a common challenge in practice. For example, when integrating datasets from multiple studies, researchers can rely on author-provided manual annotations. However, such annotations are not always available for all studies, and differences in annotation granularity across atlases can introduce inconsistencies [24,25]. In some cases, it is common to manually annotate or select a representative subset of datasets or batches to create partial high-quality labels for the full atlas [26,27]. The *partially annotated batches* scenario is thus designed to evaluate how well integration methods perform under incomplete batch-wise supervision.

Finally, the fifth and sixth partial label scenarios, *auto-annotation labels (scenario V)* and *labels with varied granularity (scenario VI)*, evaluate semi-supervised integration in two common settings: (i) labels produced by widely used automated annotation tools, and (ii) labels merged across studies that differ in annotation resolution. The growing availability of single-cell atlases and community efforts to standardize cell-type definitions provide valuable prior knowledge for cell-type identification [14]. Auto-annotation tools leverage these resources to generate labels at scale [16–18,28–30]. Although imperfect, such annotations offer a practical alternative to labor-intensive manual curation. Scenario V therefore assesses how well semi-supervised integration methods perform when provided with machine-generated labels, an increasingly relevant setting in large-scale single-cell studies. Scenario VI captures the frequent challenge of integrating datasets with mismatched label granularity. For example, one dataset may be annotated at a fine-grained level (e.g., "CD4 + Naive T," "CD8 + Memory T"), whereas another may only provide coarse labels (e.g., "T cell"). Existing benchmarks implicitly

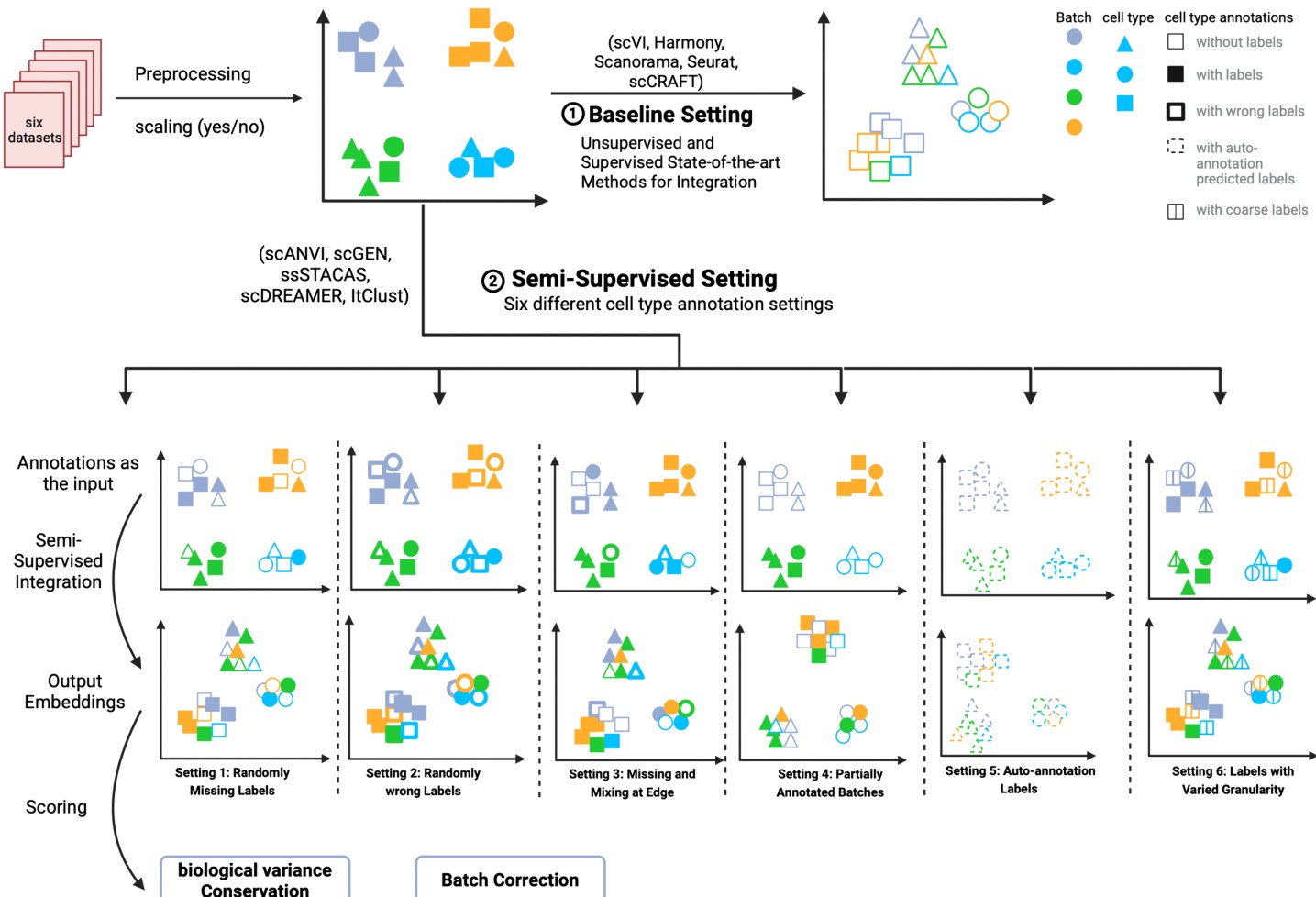

**Fig 1. Design of semi-supervised integration benchmarking.** Schematic diagram of the benchmarking workflow. In this study, ten data integration methods, including five semi-supervised algorithms and five unsupervised baselines, are evaluated across six integration datasets. Five label scenarios are designed to reflect diverse real-world conditions, including randomly missing, randomly wrong, missing and mixing at edge, partially annotated batches, automatically generated labels and labels with varied granularity, together with the baseline setting where five unsupervised methods are used and five semi-supervised method are used with full labels presented. Integration results are assessed using 9 metrics that evaluate batch effect removal and conservation of biological variance from true cell type labels (label conservation).

assume uniform label resolution, thus, overlook the need for cross-dataset label harmonization in such a realistic integration tasks.

We evaluate different methods across the above scenarios using six datasets, namely, the human pancreas, mammary epithelial cell (bct), macaque, human immune, lung atlas, and lung two species datasets, which vary in the number of cells, batches, and cell populations (cell types), in order to represent different levels of challenges in the scRNA integration task. We use the average overall performance of unsupervised methods to determine the relative difficulty of each datasets. Among the six datasets, the human pancreas dataset [31] is the easiest for unsupervised integration, containing 16,382 cells across 9 batches. This is followed by the mammary epithelial cell (bct) dataset [32–34], with 9,288 cells and 3 batches, and the macaque dataset [35], which includes 30,302 cells from 30 batches. The most challenging data set is the lung two species dataset [36], which comprises of 2 batches from human and mouse lung cells and introduces a

significant cross-species heterogeneity. The second most difficult is the lung atlas dataset [37], with 32,472 cells across 16 batches, followed by the human immune dataset [38], which contains 33,506 cells from 10 batches. Detailed preprocessing procedures and datasets descriptions are provided in Methods and Section 2 in S1 Text, respectively.

Each integration method was evaluated with nine metrics grouped into two categories: (i) conservation of biological variation and (ii) removal of batch effects (Fig 1). Biological conservation was quantified with global cluster-matching metrics—Adjusted Rand Index (ARI) [39] and normalized mutual information (NMI) [40]—together with silhouette width by cell-type label (ASW label) and cell-type LISI (cLISI) [21]. Batch-effect removal was assessed using silhouette width by batch (ASW batch) [41], the k-nearest-neighbor batch-effect test acceptance rate (kBET) [41], batch LISI (bLISI) [42], k-nearest-neighbor graph connectivity, and the true positive rate (TPR) [43]. Although TPR reflects both biological conservation and batch mixing, it is empirically more sensitive to batch mixing; following recent practice [23], we therefore treat TPR as a batch-removal metric in our summaries. Overall accuracy scores were computed as a weighted mean of the two category averages, with 60% weight on biological conservation and 40% on batch-effect removal, following [10] (see Methods). In addition these global metrics in our main discussions, we performed a targeted evaluation of rare-cell preservation on the human pancreas dataset using a rare-cell TPR (Section 9 in S1 Text).

## 2.2 Baseline setting with oracle labels

We compare five semi-supervised integration methods—scANVI, scGEN, ssSTACAS, scDREAMER, and ItClust using oracle labels, defined as the original cell-type annotations provided with each dataset. These methods are evaluated both against each other and against five unsupervised approaches—Seurat RPCA, scVI, Harmony, Scanorama, and scCRAFT to assess their label-utilization efficiency under ideal labeling conditions. In addition, the performance of the unsupervised methods serves as a baseline to reflect the intrinsic integration difficulty of the six datasets. Based on the average overall performance of these unsupervised methods, the datasets are ranked from easiest to most difficult as follows (with average unsupervised methods' overall score listed in parenthesis): human pancreas (0.7519), bct (mammary epithelial cells) (0.7496), macaque (0.7249), human immune (0.6825), lung atlas (0.6773), and lung two-species (0.6329) (Fig 2).

Beyond integration accuracy, we also evaluated the computational efficiency of these methods under the baseline setting (Section 8 in S1 Text; S1 Fig). We observed significant heterogeneity in resource requirements: while Harmony consistently demonstrated superior efficiency with minimal runtime and memory footprints, deep learning-based methods exhibited higher computational costs. Notably, scDREAMER was the slowest method overall, while ssSTACAS and Seurat showed high memory consumption on larger datasets, suggesting potential scalability constraints compared to lightweight alternatives.

scDREAMER generally achieved the highest overall weighted score among all semi-supervised methods, outperforming the strongest unsupervised competitor, scCRAFT, by an average of 7.05% across the six datasets (Fig 2a-b). Notably, scDREAMER surpassed all other semi-supervised and unsupervised integration methods across all six datasets in terms of batch removal, outperforming the second-best method, scCRAFT, by 12.18% after averaging across all datasets (Fig 2a-c). In contrast, ItClust excelled in biological conservation performance, outperforming the second-best method, scGEN, by 7.28% after averaging across all datasets.

In this ideal baseline for semi-supervised integration, the other three semi-supervised approaches also generally attain higher overall weighted scores than their unsupervised counterparts when averaged across all datasets, with robust improvement in terms of biological conservation but varied performance in batch removal (Fig 2a-c). Specifically, scANVI and scGEN both perform better than scVI in the biological conservation task, with average improvements of 6.45% and 8.04% respectively. However, considering the batch removal, while scGEN is 3.17% higher than scVI, scANVI performs worse than scVI by 5.63% on average. ssSTACAS, whose structure is similar to Seurat RPCA, shows overall similar performance to it, performing better at the macaque and lung two-species datasets but slightly worse in other four datasets. It showed a slight improvement in overall and biological conservation performance, but slightly lower batch correction performance.

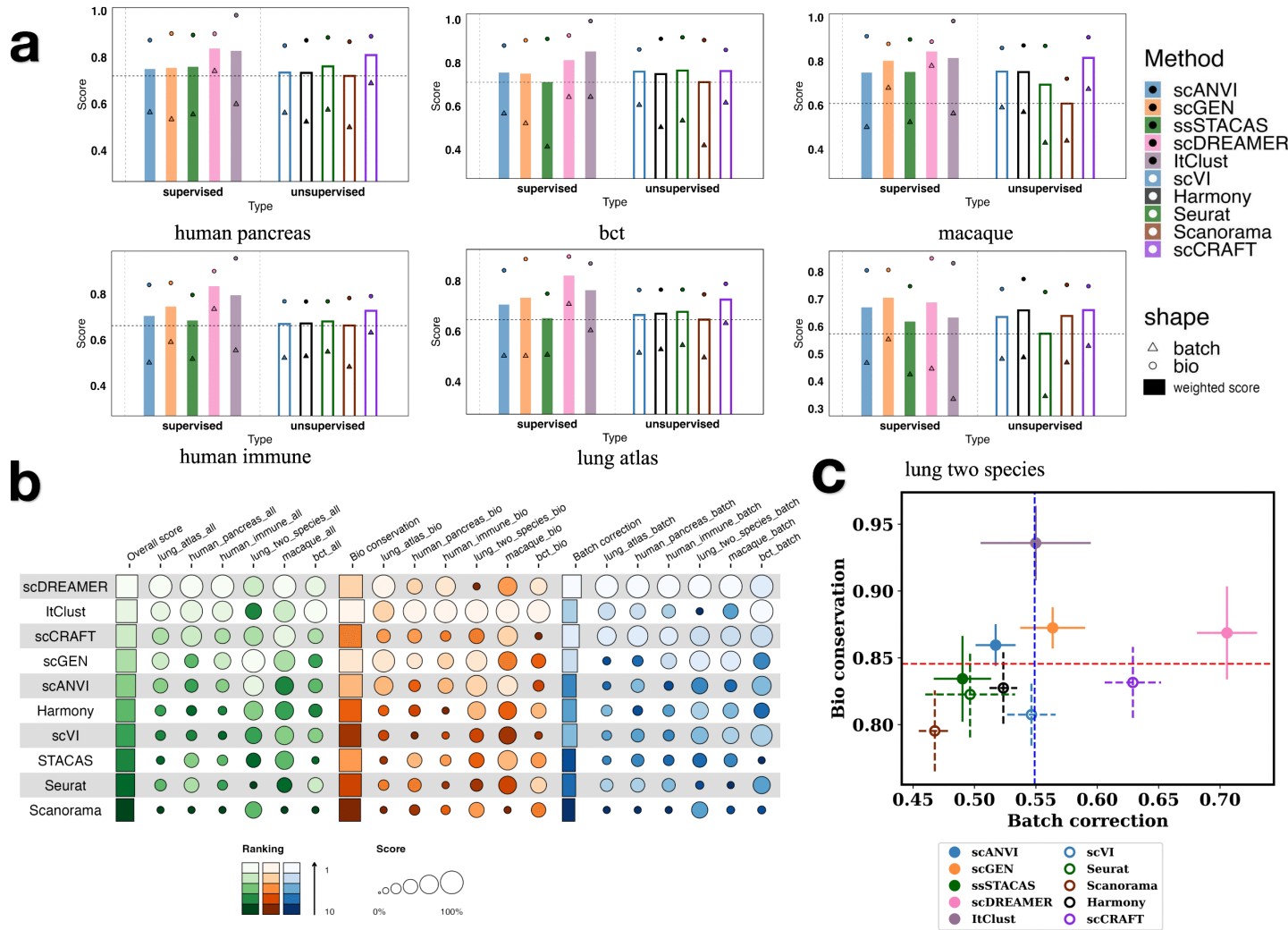

**Fig 2. Baseline settings: using unsupervised methods results.** (a) Bar plots showing the performance of all methods across six datasets under this setting. Each bar represents the overall weighted score of a method; triangles and circles indicate the batch correction and biological conservation scores, respectively. In each dataset sub-figure, the left panel displays the performance of five semi-supervised methods provided with full labels, represented by solid-colored bars. The right panel shows the performance of five unsupervised methods, depicted with empty (unfilled) bars. (b) Bubble plot of all the methods relative ranking considering the overall weighted score, bio conservation score and batch correction score. The first column of each color group represents the average performance of each method across all six datasets. (c) Scatter plot of the average overall batch correction score against the average overall bio conservation score for each methods across all datasets. The error bars represent the standard error of the mean for each method and score type across all datasets. The horizontal red dashed line represents the average bio conservation score across all methods, while the vertical blue dotted line represented the average batch correction score.

In summary, although semi-supervised approaches tend to improve biological conservation relative to their unsupervised integration backbones, their overall performance and advantage vary substantially when compared to other state-of-the unsupervised approaches. Only scDREAMER, ItClust, and scGEN performed at least as well as unsupervised methods in both biological conservation and batch removal, while showing clear improvements in at least one of these aspects (Fig 2c). When compared to scCRAFT, the strongest unsupervised method, ssSTACAS performed worse in both biological conservation and batch removal. Among the other semi-supervised approaches, although varying degrees of improvement in biological conservation were observed, only scDREAMER emerged as a clear winner. This is primarily

due to substantially lower batch removal scores observed for ItClust, scGEN, and scANVI. These observations highlight the importance of employing suitable and efficient modeling strategies for both transcriptomic variation and label information in achieving successful single-cell RNA-seq integration.

### 2.3 Scenarios I and II: Randomly missing or wrong labels

In Scenario I, we evaluated the robustness of five semi-supervised integration methods to missing annotations by randomly removing 30%, 50%, and 70% of cell-type labels in each dataset. This scenario has been widely examined previously, where semi-supervised approaches often outperform their unsupervised counterparts.

ssSTACAS and scANVI, while less effective at leveraging correct labels compared to scDREAMER, ItClust, and scGEN in the oracle setting, were robust to randomly missing labels, showing minimal performance deterioration even at 70% missingness. Both methods performed better than their respective unsupervised backbones (Seurat RPCA and scVI, respectively). In contrast, ItClust exhibited the highest sensitivity to missing labels, showing the largest drop in bio-conservation scores with merely 30% randomly missing labels and performing worse than all other unsupervised or semi-supervised approaches on every dataset except for *bct*. At 70% missingness, it consistently underperformed across all datasets (Fig 3a). scGEN and scDREAMER also showed substantial declines in performance compared to the baseline oracle setting, with deterioration increasing alongside missingness. scANVI remained the top-performing semi-supervised integration method at 70% missingness, outperforming the second-best, ssSTACAS, by an average of 3.78% in overall score across all datasets primarily due to higher bio-conservation scores, but still 3.85% lower compared the best unsupervised method, scCRAFT. However, a closer look reveals that scDREAMER achieved the best or near-best performance across most datasets, except for macaque (which has 30 batches) and lung two species (which has two batches). These exceptions contributed to scDREAMER's 5.82% lower overall score compared to scANVI. (Fig 3a).

To compare each method's overall performance across datasets while accounting for datasets difficulty, we scaled the metric scores relative to the performance of unsupervised methods for each dataset and then averaged the scaled scores across datasets for each proportion of missing labels, including performance under the oracle setting as a reference (Fig 3b). When focusing on biological signal preservation, scANVI ranked first across all levels of label missingness, outperforming the second-best method by 2.48% at 70% missingness after scaling. In contrast, for batch-effect removal, scDREAMER outperformed all other semi-supervised and unsupervised methods, exceeding the second-best method, scCRAFT, by 4.08% at 70% missingness. scCRAFT maintained a balance between the two objectives, with both scores remaining above average. Across all semi-supervised methods, performance declined in both bio-conservation and batch correction as label missingness increased, though the extent varied. Consistent with the unweighted overall performance trends, ssSTACAS and scANVI remained stable, while ItClust showed the steepest decline especially in bio-conservation. From full annotation to 30% missing labels, ItClust's overall performance dropped by 15.09%; from 30% to 70% missingness, it declined by an additional 15.79%, including a 25.28% drop in bio-conservation and 8.42% in batch correction highlighting its strong dependence on label availability.

From a closer view, we chose four datasets, human pancreas, macaque, human immune and lung atlas datasets as example, covering the relative easiest to hardest datasets, to comprehensively represent the performance of these methods under different scenario. Focusing on methods' performance on each metric (Fig 3c), we find that scDREAMER outperforms other methods in true positive rate and kBET Accept rate, surpassing the second best method scCRAFT by on average 6.84% and 23.22%, respectively across the four datasets, which contributes to its superior performance in the batch integration task. Semi-supervised methods did not show significant improvement over scCRAFT when it comes to other metrics.

In Scenario II, we assign incorrect labels to 30%, 50%, and 70% of randomly chosen cells by other cell types in a given dataset. While we observed similar trends as in Scenario I with randomly missing labels, randomly wrong labels are generally more challenging for semi-supervised integration compared to missing labels (See Section 4 in S1 Text and

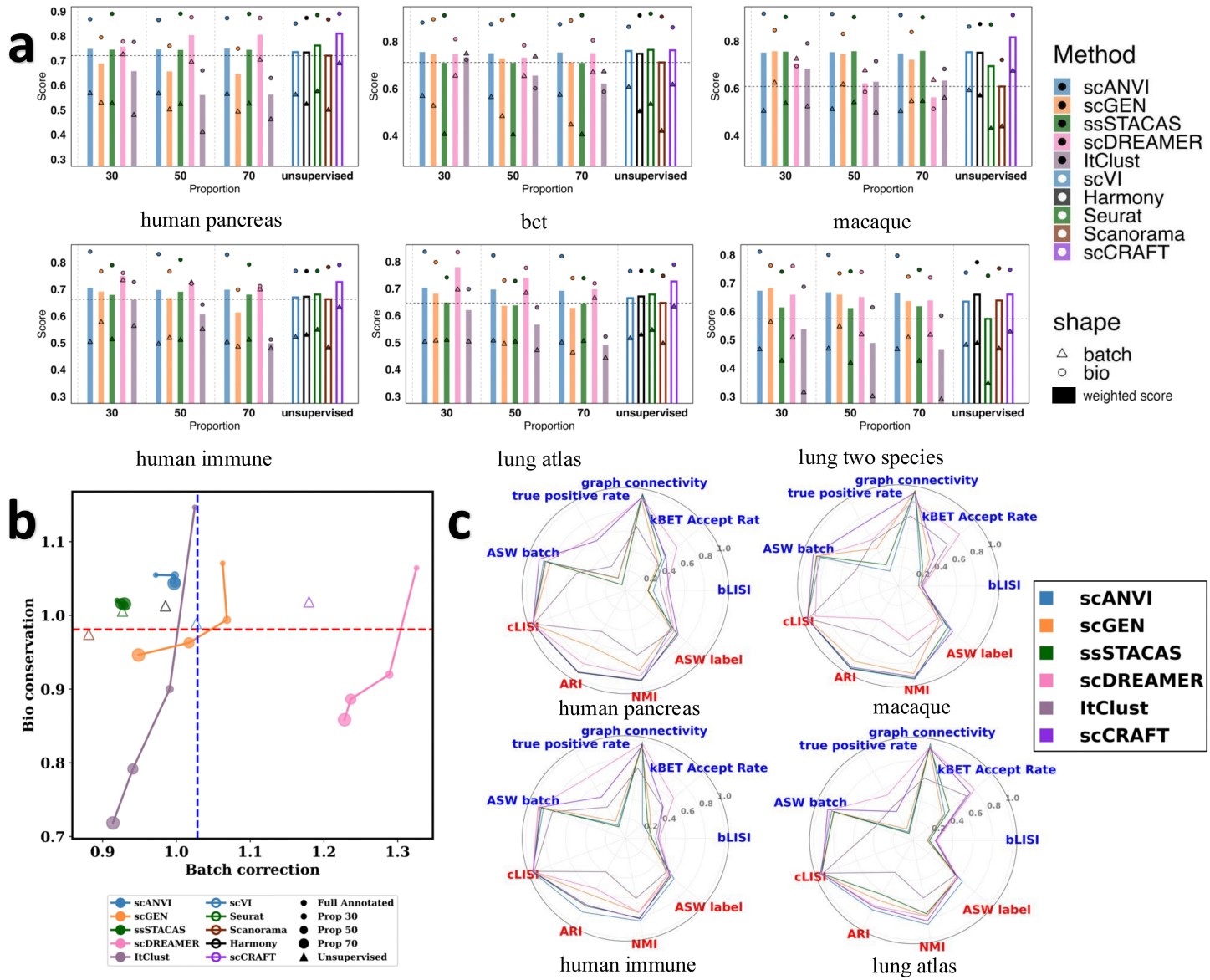

**Fig 3. Partial Label Scenario I: Randomly Missing Labels. (a)** Bar plots showing the performance of all methods across six datasets under this setting. Each bar represents the overall weighted score of a method; triangles and circles indicate the batch correction and biological conservation scores, respectively. The vertical dashed lines divide methods into four groups, namely, 30%, 50%, 70% and unsupervised approaches. The five unsupervised methods are shown on the right, represented by unfilled bars. **(b)** Scatter plot of the scaled batch correction score against the bio-conservation score for each method under the setting for different proportions, averaged across six datasets. The scaled score for each dataset and missing proportion is calculated as the ratio of overall bio-conservation/batch-mixing metric for a given method with respect to the corresponding mean using five unsupervised methods. The detailed scaling procedure can be found in Methods Section 2.2. Scaled scores for unsupervised methods are also included using unfilled triangles. Different colors indicate the methods and the size of dot shapes represent the missing proportions. The horizontal red dashed line represents the average bio conservation score across all methods (both supervised and unsupervised methods), while the vertical blue dotted line represented the average batch correction score. **(c)** Radar plots showing the performance of all methods on individual metrics for the *human pancreas, macaque, human immune,* and *lung atlas* datasets, averaged over all the three proportions for semi-supervised methods. Metrics include biological conservation (red) and batch correction (blue). As scCRAFT achieved the highest overall performance among unsupervised methods, only its scores are shown for clarity; radar plots for the remaining methods are provided in the Section 3 in S1 Text.

S2 Fig). For example, while ssSTACAS and scANVI remained two of the most robust semi-supervised methods, their performance gain over their respective unsupervised backbones (Seurat RPCA and scVI) was less significant than in the randomly missing setting, with their bio-conservation scores no longer consistently surpassing them. Moreover, under the extreme 70% wrong label condition, scDREAMER's performance on the macaque dataset drops sharply, resulting in an overall score (averaged across all datasets) 15.84% lower than scANVI's. This places scDREAMER second to last among semi-supervised methods, with scANVI remaining the top performer. In addition, scCRAFT shows better overall performance to semi-supervised methods in most datasets, surpassing scANVI and ssSTACAS by an average 5.05% and 8.20% with 70% of missingness respectively.

## 2.4 Scenario III: Missing and mixing at edge

In Scenario III, we mix labels between transcriptionally closest cell types rather than randomizing them. Concretely, in this missing and mixing at edge scenario, we first conduct unsupervised integration with Harmony, then, we examine the 30 nearest neighbors for each cell and compute the proportion of neighbors that share its true cell type. If this proportion falls below a threshold $\gamma \in [0, 1]$, the cell's label is reassigned to a different type sampled from its neighbors' label distribution. If the resampled label still matches the cell's true type, we instead mark the label as "unknown." Thus, a higher $\gamma$ results in more label changes due to more cells being selected, and once selected, the cell's label is changed to either its neighboring cell type or "unknown," making the scenario more difficult. We set the proportion $\gamma$ = 30%, 50%, 70%. This approach better reflects real-world misannotations, where prediction methods tend to assign cells to transcriptionally similar rather than arbitrary types [15].

Given the same proportion threshold, the actual percentage of reassigned cells varies across datasets, resulting in 1.08% (0.00%), 2.66% (0.88%), 3.91% (1.02%), 12.84% (4.61%), 13.63% (5.15%), and 24.96% (8.84%) reassignments for the BCT, macaque, human pancreas, lung two species, human immune, and lung atlas datasets respectively at $\gamma$ = 70%, with the proportion of missing labels reported in the parenthesis. Since human pancreas, bct and macaque have very small percents of reassignments, we focus on human immune, lung atlas and lung two species in Scenario III. The reassigned proportion in each datasets at $\gamma$ = 30%, 50%, as well as the visualization of Harmony integration results is present in Section 5 in S1 Text.

On human immune datasets, semi-supervised methods yield performance that are comparable or marginally improved relative to their unsupervised backbones. The comparison between scDREAMER and scCRAFT shows nearly equivalent batch correction score, but a 1.34% advantage for scDREAMER in the bio-conservation score when the reassignment percentage is 13.63% ($\gamma$ = 70%). The performance of scANVI and ssSTACAS was comparable with their respective unsupervised backbones, scVI and Seurat RPCA. scGEN, however, provided a more substantial 2.64% increase in overall performance compared to scVI. This general trend held in the lung two-species datasets, although in this setting, scGEN and scDREAMER both outperformed scCRAFT, by 4.09% and 0.59% (Fig 4a and c).

In contrast, within the lung atlas datasets, which exhibit the highest label reassignment percentage for each $\gamma$ threshold, a substantial performance degradation was observed across all semi-supervised methods. Even when the reassignment rate is below 30%, these methods began to underperform their corresponding unsupervised backbones. This finding suggests that existing semi-supervised approaches may possess a lower tolerance to imperfect labels than previously assumed.

Specifically, at a $\gamma$ value of 70% (reassignment rate = 24.96%), the overall scores for scANVI and scGEN fell below by 7.55% and 22.12% respectively, relative to scVI, while the ssSTACAS score was 11.81% lower than that of Seurat RPCA. Under the same conditions, scDREAMER's score was 11.40% lower than scCRAFT's, yet its performance remained comparable to or higher than other unsupervised methods. The scatter plots reveal that all semi-supervised methods exhibit a significant drop in their bio-conservation scores as the reassignment proportion increases. For example, the bio-conservation score for scANVI, a relatively stable method in the previous section, dropped by 2.48% at a 9.42%

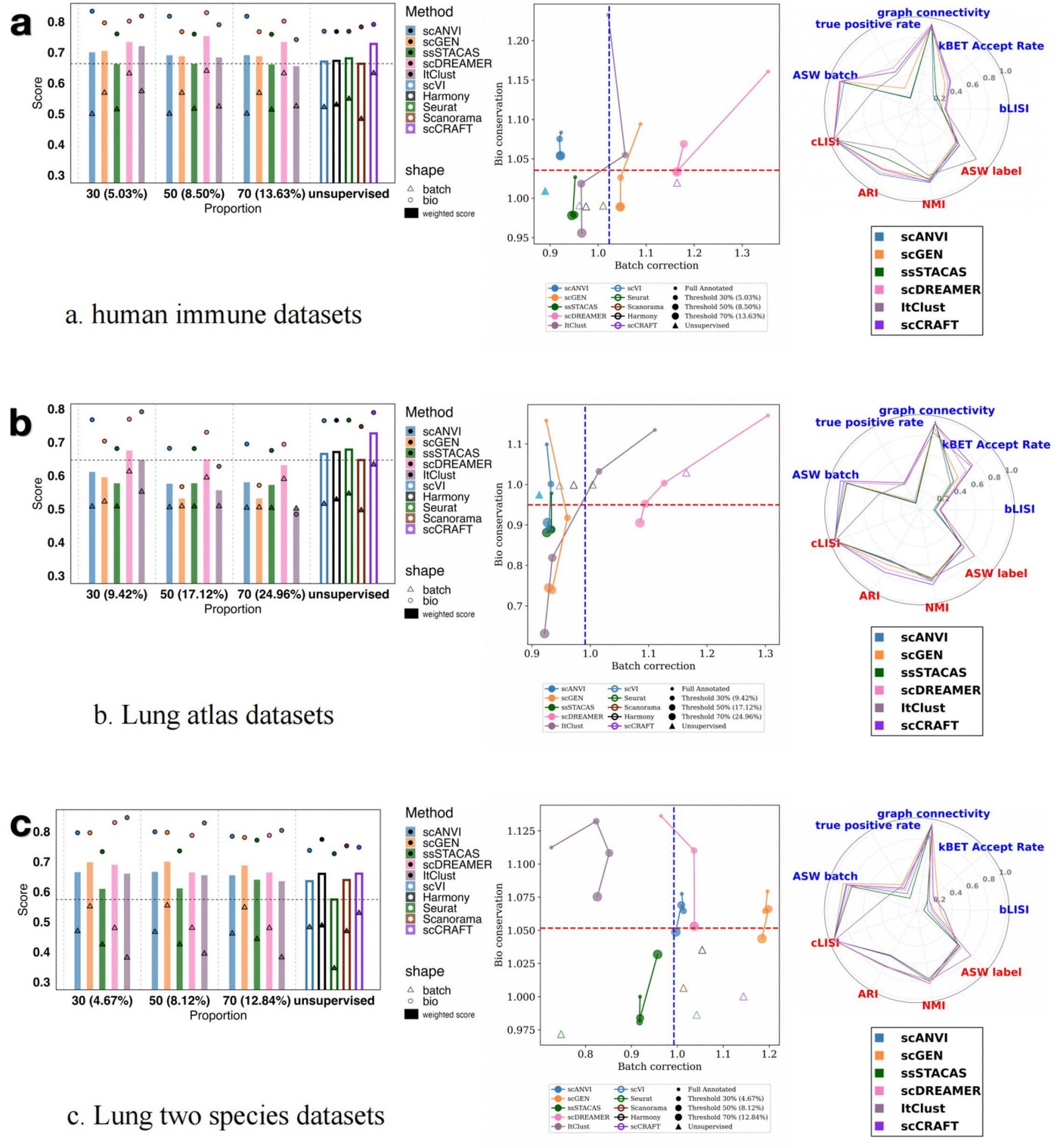

**Fig 4. Scenario III: Missing and Mixing at Edge.** Plots of semi-supervised and unsupervised methods across three selected datasets under varying levels of label reassignment in Scenario III. Results are shown for three datasets: **(a)** *human immune*, **(b)** *lung atlas*, and **(c)** *lung two species*. Each

row corresponds to a single dataset and presents a bar plot (left), a scatter plot of scaled scores (middle), and a radar plot of individual metrics (right). **Left Panels (Bar Plots):** The overall weighted score is shown for semi-supervised methods (solid bars) at different label reassignment proportions (controlled by threshold $\gamma$) and for unsupervised methods (empty bars). Overlaid circles and triangles indicate the biological conservation and batch correction scores, respectively. **Middle Panels (Scatter Plots):** The trade-off between the scaled batch correction score (x-axis) and bio-conservation score (y-axis) is visualized. Methods are color-coded, and the size of the data points corresponds to the reassignment percentage. Solid circles represent semi-supervised results at different $\gamma$ thresholds; empty triangles represent unsupervised methods. The horizontal (red) and vertical (blue) dashed lines mark the mean bio-conservation and batch correction scores, respectively. The scaling procedure is detailed in the Methods Section 2.2. **Right Panels (Radar Plots):** Performance is broken down across individual biological conservation (red) and batch correction (blue) metrics. For semi-supervised methods, scores are averaged across all proportions. For clarity, scCRAFT is shown as a representative high-performing unsupervised method. (Complete radar plots for all methods are provided in Section 3 in S1 Text).

reassignment rate ($\gamma$ = 30%) and by 2.86% at a 24.96% reassignment rate ($\gamma$ = 70%), relative to the fully annotated baseline. Although ssSTACAS's performance was stable across different thresholds, its bio-conservation score was consistently inferior to its counterpart, Seurat RPCA. The low bio-conservation scores from semi-supervised methods may be partially attributed to their weaker performance on the ASW label and ARI metrics. Notably, while scDREAMER's batch correction score also declined, it remained superior to unsupervised methods other than scCRAFT, reaffirming its advantage in this task, possibly due to its high true positive rate and kBET acceptance rate (Fig 4b).

## 2.5 Scenario IV: Integration with partially annotated batches

Scenario IV reflects a realistic use case in which users have a few well-annotated batches and aim to integrate them with newly acquired, unannotated data. We evaluate whether annotations from the current batches can improve integration performance under various semi-supervised methods. Specifically, we randomly select 30%, 50%, or 70% of the batches to lack cell-type labels (using consistent random seeds across methods and datasets for fair comparison), setting their labels to "unknown," mimicking incoming unannotated data. Other batches still include the true cell-type labels. For datasets with limited batch numbers such as lung two species (2 batches) and bct (3 batches), we adapted the setting design, and to remain consistency, we move the results for these two datasets to Section 6 in S1 Text and exclude these two datasets in the discussion below.

In this setting, scANVI and ssSTACAS are the top-performing semi-supervised methods in this setting, with nearly identical overall scores. While scANVI maintains a slight 0.23% average advantage over ssSTACAS with 70% unannotated batches, both methods significantly outperform other semi-supervised approaches. Notably, ssSTACAS surpasses the third-ranked method, scGEN, by 19.57%. Furthermore, scANVI and ssSTACAS only lag behind the top-performing method, scCRAFT, by 8.55% and 8.79% respectively. They outperform their unsupervised counterparts (scVI and Seurat RPCA) by 0.51% and 0.60% respectively under this condition. However, overall performance for semi-supervised methods did not observe significant improvement over unsupervised methods (Fig 5a).

Only scANVI and ssSTACAS showed above-average bio-conservation and batch-mixing scores with 30% of unannotated batches (Fig 5b). Similar as Scenarios I-III, semi-supervised methods showed a drastic drop in bio-conservation performance as the proportion of unannotated batches increases. For example, the bio-conservation score of ItClust, the most label-sensitive approach, falls by 49.57% when the rate of unannotated batches rises from 30% to 70%. Based on the scaled bio-conservation metric scores at 70% of missing batches annotation, scANVI is the leading semi-supervised method, outperforming ssSTACAS by 1.26% and its unsupervised counterpart, scVI, by 3.44%, while lagging just 0.68% behind the best performer, scCRAFT. Regarding batch-effect removal performance, semi-supervised methods generally performed worse than unsupervised ones even with merely 30% of unannotated batches, with only scANVI and ssSTACAS achieving results comparable to some unsupervised methods, but still much worse than the best unsupervised method, scCRAFT. For example, scANVI was surpassed by scCRAFT by 28.02% across all datasets at 30% when examining the scaled batch correction scores. Consistent with

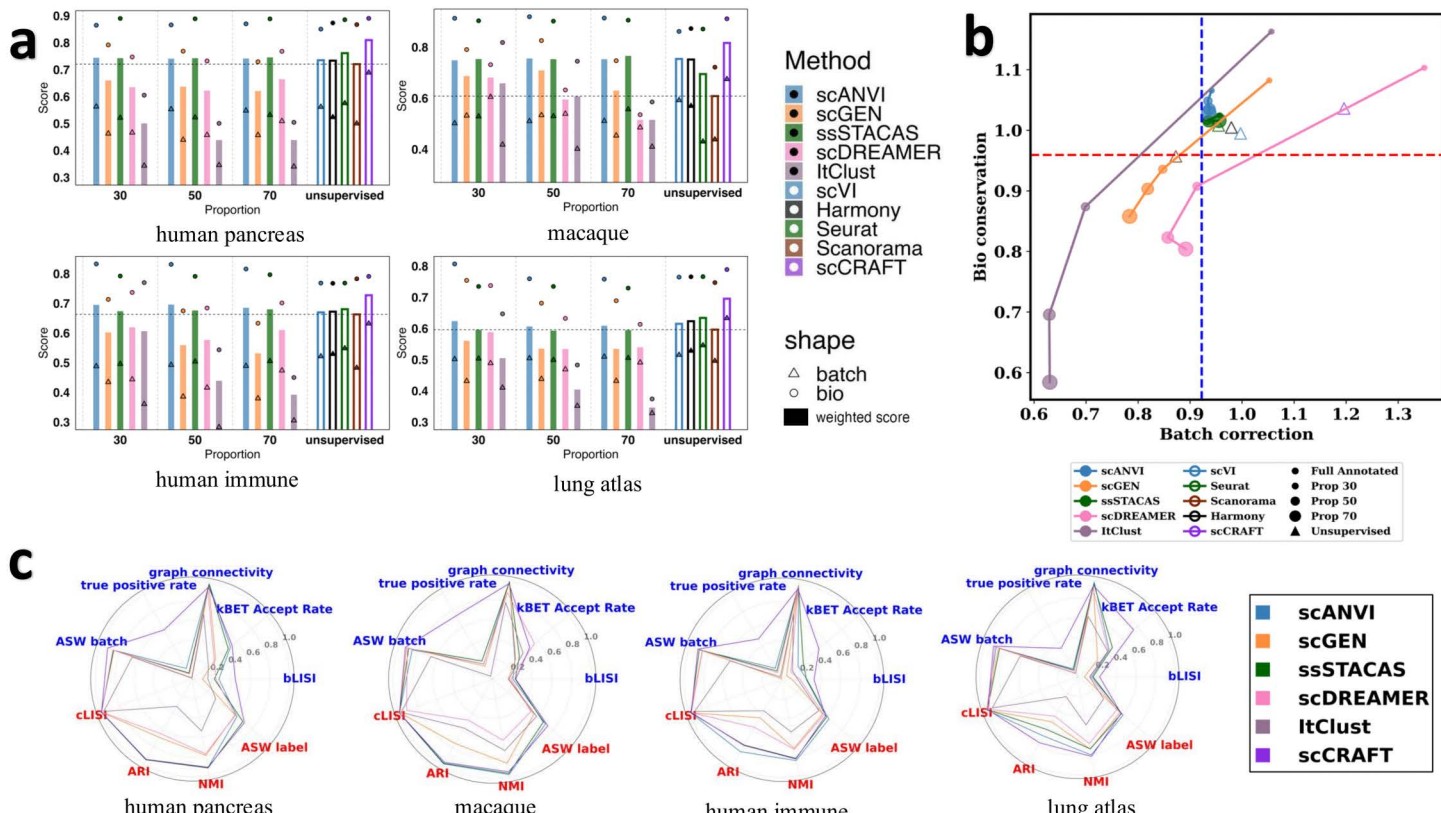

**Fig 5. Scenario IV: Partially Annotated Batches.** (**a**) Bar plots of all the methods' performance in four datasets under this setting. The bar indicates the overall weighted score of each method, while the triangle and circle represents the batch correction scores and bio-conservation scores respectively. The vertical dashed lines separate the bars into four groups, namely, 30%, 50%, 70% and unsupervised. The five unsupervised counterparts are presented on the right, depicted with empty (unfilled) bars. (**b**) Scatter plot of the scaled batch correction score against the scaled bio-conservation score for each method under the partially annotated batches setting for different proportions, averaging across four datasets. The scaled score for each dataset and missing proportion is calculated as the ratio of overall bio-conservation/batch-mixing metric for a given method with respect to the corresponding mean using five unsupervised methods. The detailed scaling procedure can be found in Methods Section 2.2. Scaled scores for unsupervised methods are also included using unfilled triangles. Different colors indicate the methods and the size of dot shapes represent the missing batch proportions. The horizontal red dashed line represents the average bio conservation score across all methods (both supervised and unsupervised methods), while the vertical blue dotted line represented the average batch correction score. (**c**) Radar plots showing the performance of all methods on individual metrics for the *human pancreas, macaque, human immune,* and *lung atlas* datasets, averaged over all the three proportions for semi-supervised methods. Metrics include biological conservation (red) and batch correction (blue). As scCRAFT achieved the highest overall performance among unsupervised methods, only its scores are shown for clarity; radar plots for the remaining methods are provided in the Section 3 in S1 Text.

the evaluations before, even though scANVI and ssSTACAS were most robust methods in Scenario IV, they did not improve over leading unsupervised approaches, with scDREAMER, ItClust and scGEN's performance deteriorating rapidly when the proportion of unannotated batches increasing and becoming worse than unsupervised methods with a moderate 30% of missing batches.

When examining individual metrics, we found that scANVI and ssSTACAS outperformed other semi-supervised methods on three of the four biological conservation metrics—ARI, ASW (label), and NMI—and were comparable to or better than the top-performing unsupervised methods on these metrics (Fig 5c). This explains their strong performance in the overall biological conservation score. However, semi-supervised methods did not show improvement over unsupervised methods on other metrics. Notably, scCRAFT demonstrated a substantial advantage in true positive rate, which largely accounts for its superior overall performance.

## 2.6 Scenario V: Integration with auto-annotated labels

As highlighted in the introduction, auto-annotated labels are readily accessible and commonly utilized as auxiliary labels for semi-supervised integration tasks [15]. Therefore, in Scenario V, we benchmark the ability of semi-supervised methods to leverage auto-annotated labels. We consider three widely used auto-annotation tools—SingleR [44], CellAssign [30], and Azimuth [29]—chosen for their popularity and practical usability [17]. CellAssign requires a marker gene matrix, whereas SingleR and Azimuth require reference datasets. Due to limitations in available references, the macaque and BCT datasets are excluded from this setting, as no suitable reference datasets could be identified for them. The predicted cell types from auto-annotation tools vary depending on the reference datasets used and often differ from the original annotations. Cell types not represented in the reference are typically labeled as "other" by CellAssign or "unknown" by SingleR. Below, we summarize the number of cell types predicted by each method across datasets (See Table 1).

Azimuth predicted labels appear to provide the best annotation quality relative to the original labels—yielding the highest overall scores across all datasets for label-sensitive methods, such as scDREAMER, ItClust, and scGEN. Among semi-supervised approaches, scDREAMER achieved the highest overall weighted score with Azimuth annotations (Fig 6a). Moreover, scDREAMER performed the best using Azimuth predicted labels among all three auto-annotation labels, achieving average improvements of 12.43% in scaled bio conservation score and 9.46% in scaled batch correction score across all datasets (Fig 6b). scDREAMER also performed favorably with CellAssign annotations, achieving the highest weighted score among semi-supervised methods in all datasets except for *lung two species*, while its performance was more mixed with SingleR. Both scANVI and ssSTACAS showed robustness to variation in auto-annotation sources (Fig 6a).

However, semi-supervised methods did not consistently outperform unsupervised alternatives (Fig 6a). For example, scCRAFT outperformed scDREAMER by an average of 5.46% in overall score, respectively, when using Azimuth annotations. scANVI performed comparably to scVI, whereas ssSTACAS performed slightly worse than Seurat RPCA in all datasets except *lung two species*, where it showed a notable improvement. The scaled metric scores relative to the performance of unsupervised methods within each dataset revealed that scANVI and ssSTACAS consistently performed at levels comparable to unsupervised approaches, with the exception of being lower than scCRAFT in terms of batch correction (Fig 6b). scGEN and ItClust showed similar bio-conservation performance to scANVI and ssSTACAS when using Azimuth predicted labels, but their performance declined with other auto-annotation sources. In terms of batch-effect removal, scCRAFT and scDREAMER remained the top two performers. Notably, scDREAMER using Azimuth annotations achieved a batch correction score comparable to scCRAFT, even surpassing it by 1.35%. However, its performance declined with other annotation sources, falling below scCRAFT in both biological conservation and batch mixing. The radar plots show that scDREAMER achieves the highest true positive rate among semi-supervised methods with average performance using three auto-annotation labels (Fig 6c). Notably, it ranks first in three datasets—except for the lung two-species dataset. However, it was surpassed by the scCRAFT in all four datasets. One important issue using these auto-annotation methods is that the predicted labels have discrepancy compared with the original ones. To assess its

**Table 1. Number of Cell Types Predicted by Auto-Annotation Tools.**

| Dataset | Original | Azimuth | CellAssign | SingleR |
|---|---|---|---|---|
| Human Pancreas | 14 | 13 | 11 | 26 |
| Human Immune | 16 | 29 | 5 | 29 |
| Lung Atlas | 17 | 20 | 8 | 36 |
| Lung Two Species | 17 | 23 | 7 | 37 |

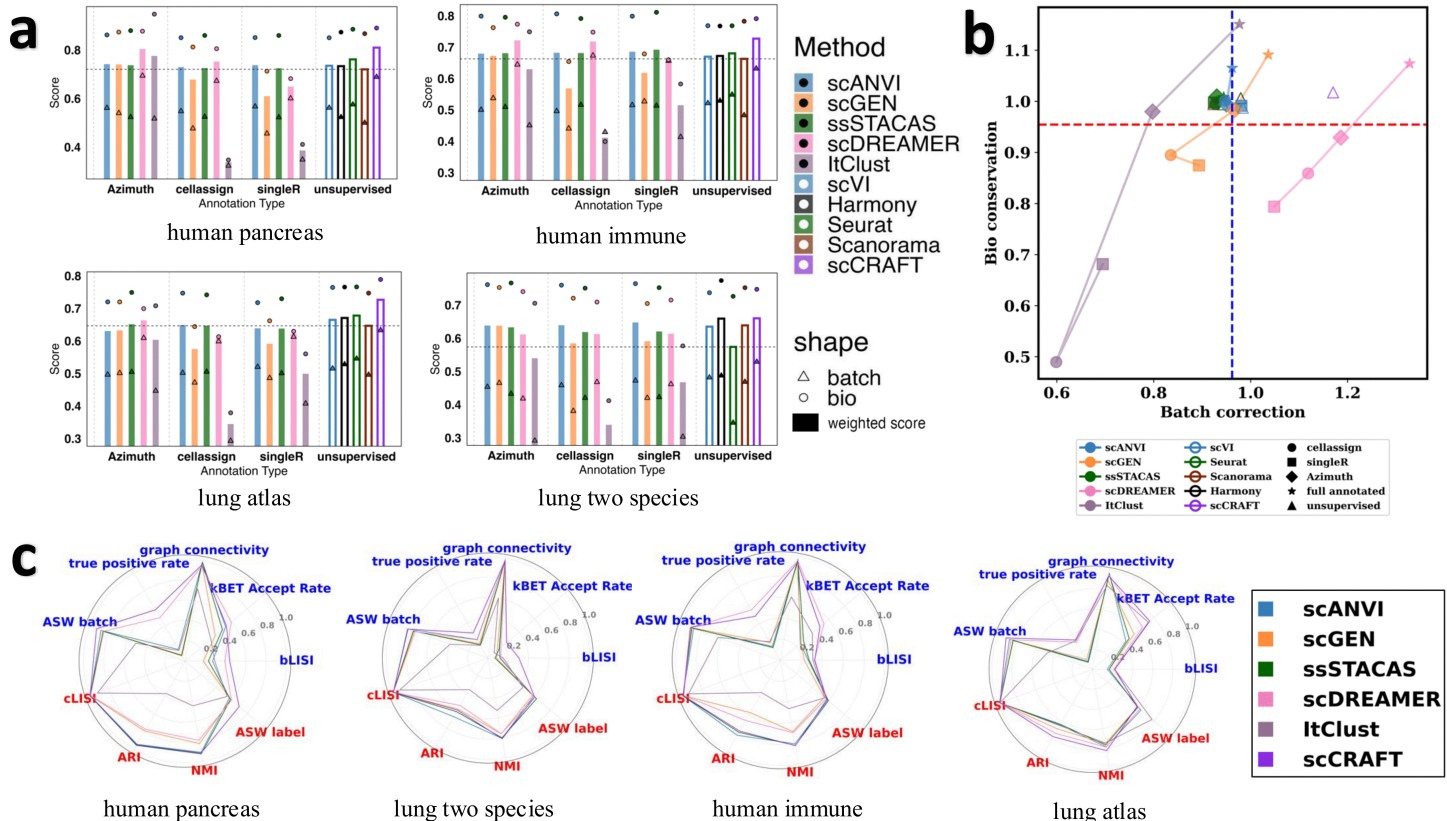

**Fig 6. Scenario V: Integration with Auto-annotated Labels.** (**a**) Bar plots showing the performance of all methods across four datasets under this setting. Each bar represents the overall weighted score of a method; triangles and circles indicate the batch correction and biological conservation scores, respectively. The vertical dashed lines divide methods into four groups: those using Azimuth, CellAssign, SingleR, and unsupervised approaches. The five unsupervised methods are shown on the right, represented by unfilled bars. (**b**) Scatter plot of scaled batch correction scores versus biological conservation scores for each method, averaged across the four applicable datasets. Different colors indicate methods, and point shapes represent the origin of the labels. The scaled score for each dataset and auto-annotated labels is calculated as the ratio of overall bio-conservation/batch-mixing metric for a given method with respect to the corresponding mean using five unsupervised methods. The detailed scaling procedure can be found in Methods Section 2.2. The horizontal red dashed line marks the average biological conservation score across all methods, while the vertical blue dotted line marks the average batch correction score. (**c**) Radar plots showing the performance of all methods on individual metrics for the *human pancreas, lung two species, human immune,* and *lung atlas* datasets, averaged over all annotation types for semi-supervised methods. Metrics include biological conservation (red) and batch correction (blue). As scCRAFT achieved the highest overall performance among unsupervised methods, only its scores are shown for clarity; radar plots for the remaining methods are provided in the Section 3 in S1 Text.

possible effect, we do fuzzy matching to compare the results which does not change our main conclusions. We include this in our Section 7.1 in S1 Text.

Recently, deep learning methods have emerged for automated cell-type annotation. To assess whether these advances improve semi-supervised integration, we tested labels generated by TOSICA, a Transformer-based tool (Section 7.2 in S1 Text). Under cross-validation (CV), TOSICA labels often matched or exceeded Azimuth, and enabled scGEN and scANVI to consistently outperform their unsupervised counterpart scVI; however, no semi-supervised method consistently surpassed scCRAFT. Importantly, the benefit of TOSICA labels was highly dependent on training scheme and dataset: CV substantially overestimated performance for scGEN and ItClust relative to external-reference training which was likely due to study-specific overfitting; and scDreamer, which was top-performing on the pancreas and human immune datasets under both training schemes, showed a marked drop on the lung atlas even with CV-based labels compared to

using classical annotation strategies. Overall, deep learning-based annotation is promising, but large-scale evaluations using strictly independent external references are still needed to establish reliability and practical utility for downstream integration.

## 2.7 Scenario VI: Integration using labels with varied granularity

Researchers frequently encounter inconsistent annotations across batches and studies, particularly when integrating datasets with varying levels of granularity. For instance, one study may provide high-resolution identities (e.g., "CD4＋Naive T," "CD8＋Memory T"), while another provides only broad lineage labels (e.g., "T cell"). To evaluate how semi-supervised methods handle such hierarchical mismatches, we utilized the hierarchical structure of the human immune dataset to simulate these discrepancies. We defined a "Coarse" schema by aggregating original "Fine" annotations into broad lineages, as detailed in Table 2.

We varied the proportion of batches with coarse annotations. Specifically, we randomly selected 30%, 50%, and 70% of the batches to be assigned with coarse labels, while the remaining batches retained their original Fine labels. We evaluated three distinct strategies for handling these mismatched labels during integration:

- Strategy I: Integrate with this inconsistent labels. Each semi-supervised method is fed with the mixed inconsistent labels (e.g., containing both "CD4+ T" and "T cell" simultaneously). This tests the model's ability to resolve hierarchical conflicts naive to the relationship between the labels.

- Strategy II: Map the fine-grained labels to coarse ones, then integrate with coarse labels. All fine-grained labels in the reference batches are downgraded to the coarse level to match the query batches. The model receives a unified, low-resolution label set.

- Strategy III: Integrate with only the fine-grained labels (Masking). The coarse labels in the query batches are treated as missing data (set to "Unknown"), while the reference batches retain fine labels. This relies on the model's ability to transfer annotations from the fine-labeled reference to the unlabeled query, similar to the Scenario IV: Partially Annotated Batches.

**Table 2. Hierarchical Mapping of Original Fine-grained Labels to Coarse-grained Lineages.**

| Coarse-grained Label | Original Fine-grained Label(s) |
| --- | --- |
| T cells | CD4＋T cells |
| | CD8＋T cells |
| | NKT cells |
| B cells | CD10＋B cells |
| | CD20＋B cells |
| | Plasma cells |
| Monocytes | CD14＋Monocytes |
| | CD16＋Monocytes |
| Dendritic cells | Monocyte-derived dendritic cells |
| | Plasmacytoid dendritic cells |
| NK cells | NK cells |
| Progenitors | HSPCs |
| | Erythroid progenitors |
| | Megakaryocyte progenitors |
| | Monocyte progenitors |
| Erythrocytes | Erythrocytes |

Regardless of the input strategy, performance was strictly evaluated against the original fine-grained ground truth to assess the preservation of biological substructures.

As illustrated in Fig 7, methods with high label dependence (scDREAMER, scGEN, and ItClust) exhibited a stepwise degradation in performance. Integration accuracy was highest with consistent coarse labels (Strategy II), declined slightly in the mixed setting (Strategy I), and reached its lowest point when coarse labels were masked (Strategy III). Crucially, performance with mixed labels remained consistently superior to the masking baseline across all batch proportions. This indicates that for these methods, while consistent supervision is ideal, "imperfect" hierarchical information is significantly more valuable than no information at all. scANVI and ssSTACAS remained robust to label granularity and label processing strategies. Their performance remained stable when provided with coarse or mixed labels, matching the results obtained using oracle labels. Like the other methods, they experienced their most notable (though small) performance declines under Strategy III as the proportion of coarse labels increased to 50% and beyond.

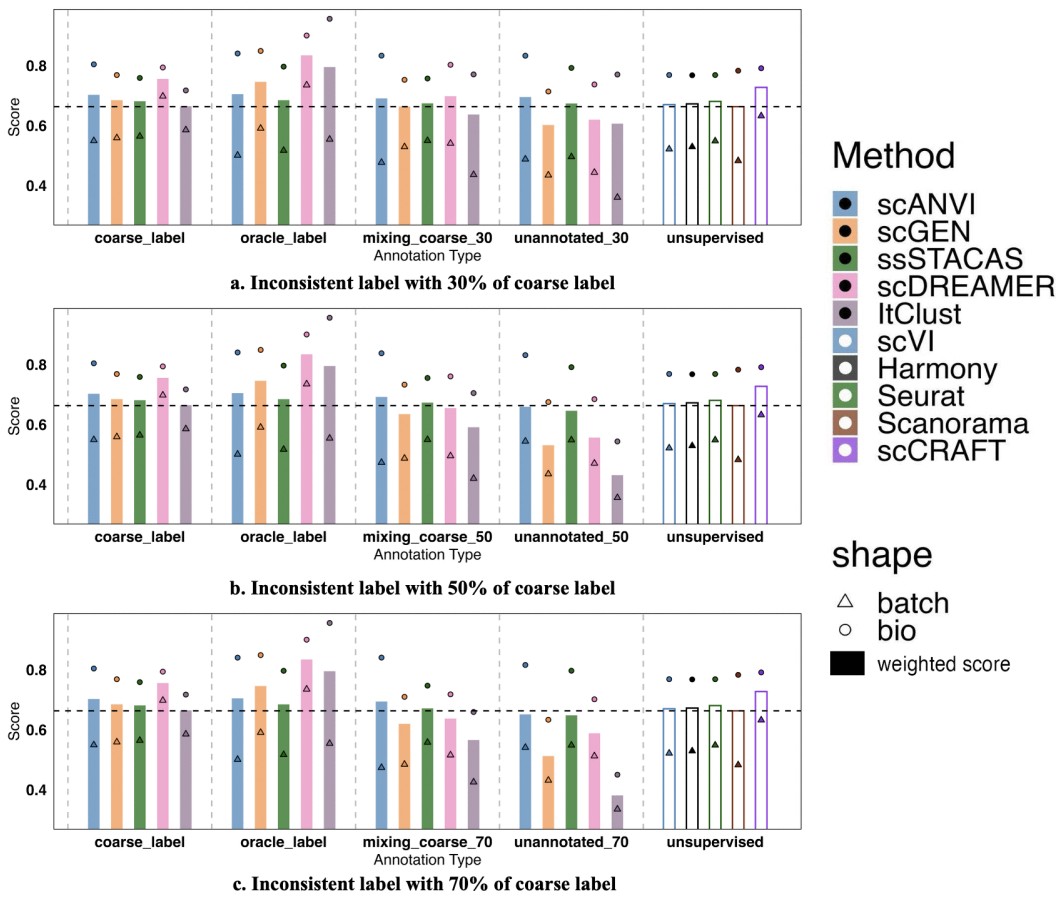

**Fig 7. Impact of inconsistent label granularity on integration performance.** (a-c) Benchmarking results on the human immune dataset where 30% (a), 50% (b), and 70% (c) of batches were assigned coarse-grained labels, while the remainder retained fine-grained labels. Bar plots show the overall integration score (y-axis) across three strategies: coarse label (harmonizing all batches to coarse levels), mixing coarse (using inconsistent labels as-is), and unannotated (masking coarse labels as "Unknown"). Semi-supervised methods' performance using oracle label and unsupervised methods' performance are also plotted as reference. Colors represent different integration methods. Note that generative methods (e.g., scDREAMER, scGEN) generally outperform the unannotated baseline when using mixed labels, whereas reference-based methods (e.g., scANVI) show sensitivity to label inconsistency. Dotted lines represent the baseline performance of unsupervised integration.).

These results provide practical guidance for integrating studies with mismatched label granularity. For label-sensitive methods, retaining all available annotations is recommended, and mapping labels to a common coarse schema may be preferable when fine-resolution labeling is infeasible for some batches. Across the semi-supervised methods examined, using mixed-granularity labels generally performed better than masking coarse labels, and robust approaches such as scANVI and ssSTACAS also exhibited higher tolerance towards inconsistent label granularity without additional preprocessing.

## 3 Conclusions

Our findings overturn the prevailing expectation that state-of-the-art semi-supervised scRNA-seq integration often outperforms unsupervised approaches and is resilient to nonexcessive imperfection in available cell labels, which has been supported by robustness checks limited to scenarios with randomly missing or erroneous labels in the literature. In contrast, our systematic evaluation reveals a more fragile reality. With perfect annotations, scDREAMER achieves the best balance between batch mixing and biological conservation, while ItClust most effectively preserves cell-type–defining signals. Yet even modest structured imperfections rapidly erode these gains: a small proportion of batch-wise missingness or lineage-boundary label swaps drives scDREAMER, ItClust, and scGEN below not only the strongest unsupervised integrator (scCRAFT) but also widely used baselines (scVI, Harmony, Seurat RPCA). The same collapse occurs when labels come from popular auto-annotation pipelines (Azimuth, CellAssign, SingleR). Only scANVI-and, to a lesser extent, ssSTACAS—maintains stable performance across all flawed-label scenarios, yet ssSTACAS seldom surpasses its unsupervised counterpart (Seurat RPCA). scANVI's overall margin over its unsupervised counterpart scVI is small, and neither method ever surpasses scCRAFT.

A distinct pattern emerges, however, when addressing inconsistent label granularity rather than outright errors. In our new scenario involving hierarchical mismatches, we found that label-sensitive generative methods (scDREAMER, scGEN, ItClust) actually benefit from retaining inconsistent annotations rather than masking them, suggesting that the structural constraints of coarse lineages are significantly more valuable to these models than no information at all. In contrast, scANVI and ssSTACAS demonstrate high robustness in this context, maintaining stable performance regardless of whether labels are coarse, mixed, or partially masked.

Although semi-supervised integration might be a promising strategy to improve integration quality by utilizing additional labeling information, unfortunately, at current stage, scCRAFT remains the most reliable "default" integrator in practice when annotation quality is uncertain. scANVI is worth considering only when a modest bump over its parent method scVI is sufficient, or when the main goal is label transfer rather than maximal integration quality. Across every scenario that involved structured label gaps or errors—batch-wise missingness, boundary mis-annotations, or labels generated by automated pipelines—none of the semi-supervised tools examined (scANVI, scGEN, ssSTACAS, scDREAMER, ItClust) delivered a clear advantage over widely used unsupervised options such as scVI, Harmony, or Seurat RPCA.

**A Practical Guide for Choosing Integration Methods.**

Based on benchmarking across the baseline setting and five scenarios, we summarize recommendations for two general use cases.

1. **High-Confidence Annotations at Coarse or Refined Resolutions:** When users possess high-quality manual annotations for all batches (e.g., baseline setting and Scenario VI with the coarse label strategy), semi-supervised methods can some times provide a distinct advantage over unsupervised approaches. In this case, we recommend scDREAMER, as it consistently achieves the highest overall scores.

2. **Noisy or Incomplete Annotations:** When annotations are moderately to highly incomplete or uncertain (e.g., missing batches, substantial boundary-region ambiguities, or imperfect auto-annotations), we recommend using a state-of-the-art unsupervised method such as scCRAFT, given the sensitivity of current semi-supervised approaches to structured

label imperfections. If users prefer established scVI- or Seurat-based workflows, scANVI and ssSTACAS are reasonable alternatives: although they rarely provide large gains, they also avoid the catastrophic failures observed in more label-dependent methods such as scGEN. Notably, despite being one of the earliest semi-supervised approaches, scANVI delivered the most consistently reliable (albeit often modest) improvements over scVI across realistic label settings in our benchmark. As auto-annotation tools improve, semi-supervised integration may increasingly benefit from machine-generated labels; however, large-scale, systematic evaluations of these emerging tools that assess annotation quality and downstream impact in practical settings are still needed before making conclusive recommendations.

**Further Discussions.**

Overall, the method-specific trends we observe can be explained by how each algorithm balances three forces: preserving manifold geometry (biological structure), enforcing batch invariance (mixing), and determining how strongly to trust and use labels. Methods whose supervision is softly coupled to a strong unsupervised backbone, or that use labels primarily as a local constraint (e.g., scANVI and ssSTACAS), tend to remain stable when labels are incomplete or imperfect because they can effectively fall back toward their unsupervised behavior as label information becomes unreliable. Yet, they fail to fully utilize the label information compared to more aggressive strategies when labels are of high qualities. In contrast, methods that use labels as a central driver of alignment or transfer (e.g., scDREAMER, ItClust, and scGEN) can perform best under oracle labels but are substantially more fragile under realistic imperfections. Importantly, structured label errors (boundary mixing, batch-specific missingness, and auto-annotation with mismatched granularity) are disproportionately harmful compared with random label corruption, because they occur precisely in ambiguous manifold regions or introduce systematic semantic mismatches that can push the embedding in a coherent but incorrect direction. This explains why top unsupervised methods remain the most reliable default when annotation quality is uncertain, and it highlights a key algorithmic need for future semi-supervised integrators: labels should be treated as uncertain signals whose influence should be adaptively down-weighted when they conflict with transcriptomics profile based cell-to-cell topology.

## 4 Methods

### 4.1 Datasets and preprocessing

We benchmarked semi-supervised data integration methods on six datasets (See Table 3). Detailed description of each datasets can be found in Section 2 in S1 Text.

Our scRNA-seq data preprocessing follows the established Scanpy [45] pipeline. Before importing the raw count matrices into Scanpy AnnData objects, we filter out low-quality cells with fewer than 300 detected genes and remove genes detected in fewer than 5 cells to avoid misleading alignments caused by dropout events or low transcriptional activity. We then store the raw counts in the count layer so that models like scVI and scANVI, which require raw count input, remain unaffected by later transformations. Next, each cell's library size is normalized to 10,000 reads, scaling gene counts by total counts per cell and adjusting them to a common scale, followed by a log transformation using $\log(1+x)$ to stabilize

**Table 3. Summary of Datasets Used in Integration.**

| Integration Task | Cell Number | Batches | Test Features |
|---|---|---|---|
| Pancreas | 16382 | 9 | Technologies, donors, rare cell types |
| Bct (mammary epithelial cells) | 9288 | 3 | Small scale with fewer cell types |
| Macaque | 30302 | 30 | Large number of batches |
| Human immune | 33506 | 10 | Tissues, laboratories, similar cell types |
| Lung atlas | 32472 | 16 | Tissues, laboratories, technologies, heterogeneity in cell composition |
| Lung two species | 20760 | 2 | 2-species integration |

variance across the dataset. After these steps, we identify the top 2,000 highly variable genes to capture the dataset's biological diversity while correcting for batch effects.

This preprocessing pipeline is uniformly applied across our benchmarking methods except for scVI, scANVI, scDREAMER and ItClust. While scVI and scANVI are configured to use the preserved raw counts, scDREAMER and ItClust employ its own self-preprocessing function.

scDREAMER's self-preprocessing function is designed to take an entire h5ad file as input and then using Scanpy to do normalization and log transformation, using the same parameters as we did. It then selects the top 2,000 highly variable genes using the Seurat RPCA method, with batch correction handled via the provided batch key. Beyond these standard steps, scDREAMER further converts the data into either a dense or sparse matrix (depending on the configuration set by sparseIP) and one-hot encodes batch information and cell type labels for downstream model use.

ItClust includes preprocessing steps, that is, filtering of cells/genes, normalization, scaling and selection of highly variables genes, similar to what we did for other methods, but with different parameters. It filters out low-quality cells with fewer than 200 detected genes and remove genes detected in fewer than 10 cells. Besides, the normalization is done to 100,000 reads, and the top 1,000 highly variable genes are selected.

Following preprocessing, we adhere to the standard benchmarking protocols for each method, with detailed procedures outlined in Section 1 in S1 Text.

## 4.2 Evaluation metrics

**4.2.1 Overview.** In our analysis of single-cell data integration methods, we employed a structured approach to evaluate performance, drawing upon established frameworks in recent studies. We focused on two critical aspects: (1) the conservation of biological variance and (2) the correction of batch effects. Following the methodology outlined in contemporary research, we selected global cluster matching (which includes the Adjusted Rand Index (ARI) and normalized mutual information (NMI), silhouette width by cell-type label (ASW label) and cLISI as four key metrics for assessing the conservation of biological variance. These metrics provide insights into how well each method preserves essential biological information and cell identity. For the correction of batch effects, our evaluation incorporated five specific metrics: the average silhouette width across batches (ASW batch), the k-nearest-neighbor batch effect test (kBET Accept Rate), bLISI, k-nearest-neighbor graph connectivity, and true positive rate. These metrics collectively assess the effectiveness of each integration method in harmonizing data across different batches, ensuring that technical variations do not obscure biological signals. Overall accuracy scores were computed by taking the weighted mean of all metrics computed for an integration run, with a 60/40 weighting of biological variance conservation (bio-conservation) to batch effect removal irrespective of the number of metrics computed.

**4.2.2 Scaling procedure for scatter plots.** In Figs 3b, 4b, 5b, and 6b, we report scaled scores for both bio-conservation and batch correction to facilitate comparison across datasets with varying baseline difficulties. The scaling procedure is defined as follows:

Let $R_{m,d}^{(k)}$ denote the raw score for method $m$ on dataset $d$ for metric type $k$ (where $k \in \{\mathrm{Bio}, \mathrm{Batch}\}$). Let $\mathcal{U}$ represent the set of unsupervised methods serving as the baseline group.

**1. Compute Unsupervised Baseline:** For each dataset $d$ and metric $k$, we calculate the baseline $B_d^{(k)}$ as the mean raw score of all unsupervised methods:

$$B_d^{(k)} = \frac{1}{|\mathcal{U}|} \sum_{u \in \mathcal{U}} R_{u,d}^{(k)}$$

(1)

**2. Normalize Raw Scores:** We normalize the raw score of every method $m$ (including both supervised and unsupervised methods) by the dataset-specific baseline:

$$S_{m,d}^{(k)} = \frac{R_{m,d}^{(k)}}{B_d^{(k)}}$$

(2)

Consequently, a scaled score of $S = 1$ represents performance equivalent to the *average* unsupervised method. Individual unsupervised methods may exhibit scaled scores deviating from 1 (i.e., $S_{u,d}^{(k)} \neq 1$) if they perform better or worse than the group mean.

**3. Average Scaled Scores:** Finally, to summarize performance across all $D$ datasets for a specific labeled data proportion, we compute the mean scaled score:

$$\bar{S}_m^{(k)} = \frac{1}{D} \sum_{d=1}^{D} S_{m,d}^{(k)}$$

(3)

### 4.2.3 Metrics for Bio-conservation.

**4.2.3.1 ARI:** The adjusted Rand index (ARI) measures agreement between two partitions by adjusting the raw Rand index for chance overlap. In our workflow, we compared true cell-type annotations against Louvain clusters derived from the integrated scRNA-seq data. By accounting for random label assignments, the ARI yields 0 for clusterings no better than chance and 1 for perfect concordance. We computed ARI using the scib package.

**4.2.3.2 NMI:** The normalized mutual information (NMI) quantifies the similarity between two clusterings by measuring the amount of shared information, normalized to account for differences in cluster sizes. We used NMI to evaluate how well Louvain clusters derived from the integrated data align with known cell-type annotations. An NMI of 0 indicates no mutual information (random agreement), while a value of 1 reflects perfect overlap. All NMI scores were calculated using the scib package.

**4.2.3.3 Cell type Average Silhouette Width (ASW label):** The average silhouette width (ASW) quantifies how well-defined clusters are by comparing the average within-cluster distance of a cell to its average distance from the nearest neighboring cluster. Scores range from -1–1, with higher values indicating more distinct and cohesive clusters.

$$\text{ASW} = \frac{1}{N} \sum_{i=1}^{N} \frac{b(i) - a(i)}{\max\{a(i), b(i)\}}$$

To evaluate biological conservation, we computed ASW based on known cell-type labels (ASW label) using the integrated embedding. An ASW near 1 suggests strong preservation of cell-type structure, whereas values near 0 or negative reflect overlapping or misclassified clusters. To facilitate comparison across methods, we linearly scaled the ASW label scores to a 0–1 range using the formula:

$$\text{ASW}_{\text{label}} = \frac{\text{ASW} + 1}{2}$$

We compute this metric using scib package.

**4.2.3.4 Cell Type Local Inverse Simpson's Index (cLISI):** The cell-type Local Inverse Simpson's Index (cLISI) quantifies the diversity of cell types within local neighborhoods to assess biological variance preservation in integrated data. It is computed by applying the inverse Simpson's Index to the distribution of cell-type labels among each cell's nearest neighbors, capturing the effective number of distinct cell types present locally. Letting $B$ represents the number of unique cell types, and $p_b$ denote the probability of observing cell type $b$ in a neighborhood, cLISI is defined as:

$$\text{cLISI} = 1 - \frac{\sum_{b=1}^{B} p(b)^2 - 1}{B - 1}$$

The final cLISI score is averaged across all cells and then min-max scaled to the range [0, 1], where higher values reflect better conservation of biological diversity. This metric is particularly useful for detecting whether cell-type structure is preserved locally after integration.

**4.2.4 Metrics for batch-effect removal. 4.2.4.1 Average Silhouette Width for Batches (ASW batch):** The batch average silhouette width (ASW batch) assesses how well batches are mixed within each biological cell type after integration. It is based on the absolute silhouette width computed on batch labels, where a value close to 0 indicates good batch mixing, and larger values reflect residual batch effects. To ensure that higher scores correspond to better integration, the silhouette values are subtracted from 1. Specifically, the batchASW for each cell type is computed by averaging $1 - |s(i)|$, where $s(i)$ is the silhouette score of cell $i$ with respect to batch labels. These cell-type–specific scores are then averaged across all cell types to obtain the final batch ASW:

$$\text{batch ASW}_j = \frac{1}{|C_j|} \sum_{i \in C_j} \left(1 - s_{\text{batch}}(i)\right)$$

$$\text{batch ASW} = \frac{1}{|M|} \sum_{j \in M} \text{batch ASW}_j$$

where $C_j$ is the set of cells of cell type $j$, and $M$ is the set of all unique cell types. A final batch ASW score close to 1 indicates ideal batch mixing, while a score near 0 implies poor integration. We compute this metric using scib package.

**4.2.4.2 k-Nearest Neighbor Batch Effect Test (kBET):** The k-nearest neighbor Batch Effect Test (kBET) evaluates local batch label composition against the global batch distribution to assess batch mixing. The test is applied iteratively to randomly sampled subsets of cells, with a chi-squared test determining whether local neighborhoods deviate significantly from expected label proportions. For each embedding or corrected feature space, k-nearest neighbor graphs are constructed with a fixed neighborhood size to ensure consistency across methods. When applied, kBET accounts for technical variation by testing batch-wise and adapting the neighborhood size (typically between 10 and 100) based on local connectivity. For disconnected graphs, the test is limited to connected components; if more than 25% of cells belong to components too small for testing, a default score of 1 is assigned, indicating inadequate correction. The final kBET score is computed as the inverse of the average rejection rate across tested cells, where values near 1 indicate well-mixed batches and values near 0 reflect poor integration. All kBET scores were computed using the kBET R package (v.0.99.6).

**4.2.4.3 Batch Label Identity Score Index (bLISI):** The batch Local Inverse Simpson's Index (bLISI) assesses batch mixing by quantifying the diversity of batch labels within each cell's local neighborhood. A higher bLISI score indicates that a cell's neighbors are drawn evenly from multiple batches, reflecting effective batch correction. Ideally, well-integrated datasets yield bLISI values approaching the total number of batches, suggesting uniform local batch composition. To normalize the score across datasets with varying batch counts, bLISI is scaled using the transformation:

$$\text{bLISI}_{\text{scaled}} = \frac{\text{bLISI} - 1}{B - 1}$$

where $B$ is the total number of batches. This scaling bounds the score between 0 and 1, where values near 1 indicate optimal batch mixing and values near 0 suggest poor integration.

**4.2.4.4 Graph connectivity:** Graph connectivity evaluates whether cells of the same biological identity remain connected in the integrated k-nearest neighbor (kNN) graph, serving as an indicator of biological conservation. For each cell type, the method constructs a subgraph from the global kNN graph by retaining only cells of that type, and then computes the size of the largest connected component relative to the total number of cells of that type. The final graph connectivity score is obtained by averaging these proportions across all cell types:

$$GC = \frac{1}{|C|} \sum_{c \in C} \frac{|LCC(G(N_c; E_c)|}{|N_c|}$$

where $C$ represents the set of cell identity labels, $|LCC()|$ is the number of nodes in the largest connected component of the graph and $|N_c|$ is the number of nodes with cell identity $c$. A score close to 1 indicates that cells of the same type are well-connected post-integration, reflecting strong preservation of biological structure.

**4.2.4.5 Proportion of True Positive Cells (True Positive Rate):** The true positive rate (TPR) evaluates integration quality at the single-cell level by assessing the local consistency of cell type and batch composition. A cell is considered positive if its local neighborhood consists predominantly of cells from the same annotated type. Among these, true positives are those whose local batch distribution also reflects the global batch proportions, indicating proper batch mixing. The TPR is then defined as the fraction of true positives among all positive cells, capturing both biological variance conservation and batch effect correction. Higher TPR values signify that the integration method preserves cell identity while achieving balanced batch mixing in local neighborhoods.

**4.2.5 Overall scores and metric aggregation.** Metrics were run on the integrated embeddings output by each method. The overall score, $S_{overall,i}$, for each integration run $i$ was calculated by taking the weighted mean of the batch removal score, $S_{batch,i}$, and the bio-conservation score, $S_{bio,i}$, following the equation:

$$S_{overall,i} = 0.6 \times S_{bio,i} + 0.4 \times S_{batch,i}$$

Here we assign higher weights to bio-conservation score since we focus more on semi-supervised methods' performance on the preservation of biological variations.

Specifically, the partial scores were computed by averaging all raw metrics that belong to each type via:

$$S_{batch,i} = \frac{1}{|M_{batch}|} \sum_{m \in M_{batch}} m(X_i)$$

and

$$S_{bio,i} = \frac{1}{|M_{bio}|} \sum_{m \in M_{bio}} m(X_i)$$

Here, $X_i$ denotes the integration output for run $i$, and $M_{batch}$ and $M_{bio}$ denote the set of metrics belong to batch removal and bio-conservation scores respectively.

## Supporting information

**S1 Text. Supplementary Note.** This file contains detailed descriptions of integration methods, datasets, and additional benchmarking results.
(PDF)

**S1 Fig. Computational Performance of Integration Methods across All Datasets Under the Baseline Setting.** The scatter plots depict the trade-off between computation time (*y*-axis, measured in seconds) and peak CPU memory usage (*x*-axis, measured in gigabytes). The figure is divided into six panels (a–f), each corresponding to a specific dataset: (a) human pancreas, (b) bct, (c) macaque, (d) human immune, (e) lung atlas, and (f) lung two species. Each dot represents one of the ten evaluated integration methods. Unsupervised methods are represented by empty dots, while semi-supervised or supervised methods are indicated by filled dots. Notably, the reported computation time for scANVI includes the training time of the scVI model, as scANVI relies on a pre-trained scVI model for initialization. Methods located closer to the origin (bottom-left corner) of each plot demonstrate higher computational efficiency.
(TIF)

**S2 Fig. Partial Label Scenario II: Randomly Wrong Labels.** Bar plots showing the performance of all methods across six datasets under this setting. Each bar represents the overall weighted score of a method; triangles and circles indicate the batch correction and biological conservation scores, respectively. The vertical dashed lines divide methods into four groups, namely, 30%, 50%, 70% and unsupervised approaches. The five unsupervised methods are shown on the right, represented by unfilled bars. Scatter plot of the scaled batch correction score against the bio-conservation score for each method under the setting for different proportions, averaged across six datasets. The scaled score for each dataset and missing proportion is calculated as the ratio of overall bio-conservation/batch-mixing metric for a given method with respect to the corresponding mean using five unsupervised methods. Scaled scores for unsupervised methods are also included using unfilled triangles. Different colors indicate the methods and the size of dot shapes represent the missing proportions. The horizontal red dashed line represents the average bio conservation score across all methods (both supervised and unsupervised methods), while the vertical blue dotted line represented the average batch correction score. Radar plots showing the performance of all methods on individual metrics for the *human pancreas, macaque, human immune,* and *lung atlas* datasets, averaged over all the three proportions for semi-supervised methods. Metrics include biological conservation (red) and batch correction (blue). As scCRAFT achieved the highest overall performance among unsupervised methods, only its scores are shown for clarity; radar plots for the remaining methods are provided in Section 5 in S1 Text.
(TIF)

**S3 Fig. Radar Plots for Randomly Missing Setting.** Radar plot of all the methods' performance on each metric in human pancreas, macaque, human immune and lung atlas datasets under the randomly missing setting, averaging across all the three proportions, where the metrics include bio-conservation metrics (shown in red) and batch correction metrics (shown in blue).
(TIF)

**S4 Fig. Radar Plots for Randomly Wrong Setting.** Radar plot of all the methods' performance on each metric in human pancreas, macaque, human immune and lung atlas datasets under the randomly wrong setting, averaging across all the three proportions, where the metrics include bio-conservation metrics (shown in red) and batch correction metrics (shown in blue).
(TIF)

**S5 Fig. Radar Plots for Partially Annotated Batches Setting.** Radar plot of all the methods' performance on each metric in human pancreas, macaque, human immune and lung atlas datasets under the partially annotated batches setting, averaging across all the three proportions, where the metrics include bio-conservation metrics (shown in red) and batch correction metrics (shown in blue).
(TIF)

**S6 Fig. Radar Plots for Missing and Mixing at Edge Setting.** Radar plot of all the methods' performance on each metric in human pancreas, macaque, human immune and lung atlas datasets under the missing and mixing at edge setting,

averaging across all the three proportions, where the metrics include bio-conservation metrics (shown in red) and batch correction metrics (shown in blue).
(TIF)

**S7 Fig. Radar Plots for Auto-annotation Labels Setting.** Radar plot of all the methods' performance on each metric in human pancreas, lung two species, human immune and lung atlas datasets under the auto-annotation labels setting, averaging across all the three proportions, where the metrics include bio-conservation metrics (shown in red) and batch correction metrics (shown in blue).
(TIF)

**S8 Fig. Harmony Clustering Results for each datasets in Missing and Mixing at Edge Setting.** The plot shows Harmony clustering for five datasets (subpanels a–f: bct, macaque, human pancreas, lung two species, human immune, and lung atlas). In each subpanel, the three side-by-side plots correspond to the three predefined proportion thresholds. Within every plot, cells are colored by their original cell type labels to preserve biological context, while black points mark missing or mixed assignments. The label beneath each plot gives the proportion threshold value, with the mismatch fraction (mismatched cells/total cells) in parentheses.
(TIF)

**S9 Fig. Bar plots for lung two species and bct datasets in Partially Annotated Batches Setting.** Bar plots showing the performance of all methods (ssSTACAS was excluded due to its design) across lung two species and bct datasets under this setting. Each bar represents the overall weighted score of a method; triangles and circles indicate the batch correction and biological conservation scores, respectively. The vertical dashed lines divide methods into three groups. Results for BCT (three batches) are presented with 1 or 2 unannotated batches. Results for lung two-species (two batches) are presented with either zero or one unannotated batch. The five unsupervised methods are shown on the right, represented by unfilled bars.
(TIF)

**S10 Fig. Heatmap for Difference between Fuzzy Matching Labels and Raw Azimuth Labels.** This plot presents the differences in a grid format. Rows represent the Methods, and columns represent metrics. The color intensity and hue of each cell directly visualize the calculated difference.
(TIF)

**S11 Fig. Bar Chart for Difference between Fuzzy Matching Labels and Raw Azimuth Labels.** For each 'Metrics', this chart displays a group of bars. Each bar in the group corresponds to one of the methods. The height of the bar indicates the difference in scores between the two files.
(TIF)

**S12 Fig. Integration with Auto-annotated Labels Generated By TOSICA.** Bar plots showing the performance of all methods across four datasets under this setting. Each bar represents the overall weighted score of a method; triangles and circles indicate the batch correction and biological conservation scores, respectively. The vertical dashed lines divide methods into four groups: those using Azimuth, CellAssign, SingleR, TOSICA and unsupervised approaches. The five unsupervised methods are shown on the right, represented by unfilled bars. Detailed performance comparison specifically for the human immune dataset under two different auto-annotation strategies: Integration performance using TOSICA labels generated via the 5-fold cross-validation strategy (internal prediction). Integration performance using TOSICA labels predicted based on the external CITE-seq reference dataset. Scatter plot of scaled batch correction scores versus biological conservation scores for each method, averaged across the three applicable datasets, and for using cross validation generated labels only. Different colors indicate methods, and point shapes represent the origin of the labels. The scaled

score for each dataset and auto-annotated labels is calculated as the ratio of overall bio-conservation/batch-mixing metric for a given method with respect to the corresponding mean using five unsupervised methods. The horizontal red dashed line marks the average biological conservation score across all methods, while the vertical blue dotted line marks the average batch correction score.
(TIF)

**S13 Fig. Evaluation of Rare Cell Type Preservation.** Performance on rare cell-type preservation of integration methods on the human pancreas dataset. Bar plot showing the absolute cell count for each cell type and the distribution of batches across different cell types. Box plots displaying the distribution of TPR scores for each method across all settings, illustrating the median performance and stability. The x-axis displays the integration methods, and the y-axis displays the true positive rate (TPR). The vertical dashed line separates semi-supervised methods (left) from unsupervised methods (right). Bar charts depicting performance across five experimental scenarios. The x-axis displays the evaluated methods, and the y-axis displays the average true positive rate (TPR). Each subplot depicts one scenario in the results. (1) Baseline performance with full annotations. (2–5) Robustness analysis under varying degrees of label noise: randomly missing labels (2), randomly wrong labels (3), and partially annotated batches (4). Data points represent the average TPR across all tested proportions (30%, 50%, 70%). (5) Performance using automated annotations (averaged across Azimuth, CellAssign, and SingleR). The dashed line separates between supervised and unsupervised methods within each subplot.
(TIF)

**S1 Table. Proportion of Reassigned Cells.** The table reports the detailed reassigned proportion in each dataset at the threshold of $\gamma = 30\%$, 50% and 70%. The proportion of cells whose labels were set to missing ("Unknown") is also reported in the parenthesis. Datasets include BCT, Macaque, Human Pancreas, Lung Two Species, Human Immune, and Lung Atlas.
(PDF)

**S2 Table. Mapping of Given Cell Types to Azimuth Predicted Cell Types.** This table shows the mapping of given cell types to Azimuth predicted cell types for the human immune dataset. Azimuth predicted more kinds of cell types than the given labels, so we mapped the Azimuth labels to the given labels based on our prior knowledge of cell types. The table includes mappings for various cell types including CD4 + T cells, CD8 + T cells, CD14 + Monocytes, CD16 + Monocytes, CD20 + B cells, Erythrocytes, HSPCs, Megakaryocyte progenitors, Monocyte-derived dendritic cells, NK cells, NKT cells, Plasma cells, and Plasmacytoid dendritic cells.
(PDF)

**S3 Table. Rank Comparisons for Bio Conservation, Batch Correction, and Overall Weighted Scores.** This table presents rank comparisons for bio conservation, batch correction, and overall weighted scores comparing fuzzy matching labels versus raw Azimuth labels. The table shows rankings for five methods (scANVI, scGEN, ssSTACAS, scDREAMER, and ItClust) across three evaluation categories: Bio Conservation, Batch Correction, and Overall Weighted, with separate columns for Fuzzy and Raw label comparisons.
(PDF)

## Author contributions

**Conceptualization:** Xiaoyu Shen, Chuan He, Leying Guan.

**Data curation:** Xiaoyu Shen.

**Formal analysis:** Xiaoyu Shen.

**Investigation:** Xiaoyu Shen.

**Methodology:** Xiaoyu Shen.

**Software:** Xiaoyu Shen.

**Supervision:** Leying Guan.

**Visualization:** Xiaoyu Shen.

**Writing – original draft:** Xiaoyu Shen.

**Writing – review & editing:** Chuan He, Leying Guan.

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
