## [Decision Letter · Decision Letter 0]

16 Nov 2025

A Benchmark of Semi-Supervised scRNA-seq Integration Methods in Real-World Scenarios

PLOS Computational Biology

Dear Dr. He,

Thank you for submitting your manuscript to PLOS Computational Biology. After careful consideration, we feel that it has merit but does not fully meet PLOS Computational Biology's publication criteria as it currently stands. Therefore, we invite you to submit a revised version of the manuscript that addresses the points raised during the review process.

We look forward to receiving your revised manuscript.

Kind regards,

Tao Wang, Ph.D

Academic Editor

PLOS Computational Biology

Ferhat Ay

Section Editor

PLOS Computational Biology

**Additional Editor Comments:**

The study addresses an important topic and is based on a well-designed benchmarking framework. However, all reviewers identified several major issues that must be addressed before the manuscript can be considered for publication.

In particular, please:

1. Include a new scenario addressing inconsistent label schemas across datasets.

2. Provide deeper analysis explaining why specific methods perform differently under varying conditions.

3. Expand the discussion and conclusion to offer actionable guidance for future method development.

4. Incorporate recent deep learning-based auto-annotation tools and corresponding analyses.

5. Report computational efficiency (runtime and memory) and clarify metric calculations.

Overall, the manuscript has strong potential, but substantial revision and additional analyses are necessary. Please submit a detailed, point-by-point response with your revised manuscript.

**Journal Requirements:**

At this stage, the following Authors/Authors require contributions: leying Guan, Chuan He, and Xiaoyu Shen. Please ensure that the full contributions of each author are acknowledged in the "Add/Edit/Remove Authors" section of our submission form.

4) Your manuscript is missing the following sections: Discussion.  Please ensure all required sections are present and in the correct order. Make sure section heading levels are clearly indicated in the manuscript text, and limit sub-sections to 3 heading levels. An outline of the required sections can be consulted in our submission guidelines here:

5) Please upload all main figures as separate Figure files in .tif or .eps format. For more information about how to convert and format your figure files please see our guidelines:

6) We have noticed that you have uploaded Supporting Information files, but you have not included a list of legends. Please add a full list of legends for your Supporting Information files after the references list.

7) Please provide a detailed Financial Disclosure statement. This is published with the article. It must therefore be completed in full sentences and contain the exact wording you wish to be published.

1) Please clarify all sources of financial support for your study. List the grants, grant numbers, and organizations that funded your study, including funding received from your institution. Please note that suppliers of material support, including research materials, should be recognized in the Acknowledgements section rather than in the Financial Disclosure

2) State the initials, alongside each funding source, of each author to receive each grant. For example: "This work was supported by the National Institutes of Health (####### to AM; ###### to CJ) and the National Science Foundation (###### to AM)."

3) State what role the funders took in the study. If the funders had no role in your study, please state: "The funders had no role in study design, data collection and analysis, decision to publish, or preparation of the manuscript."

4) If any authors received a salary from any of your funders, please state which authors and which funders..

8) Your current Financial Disclosure states, "The author(s) received no specific funding for this work.".

However, your funding information on the submission form indicates receiving funds.

Please indicate by return email the full and correct funding information for your study and confirm the order in which funding contributions should appear. Please be sure to indicate whether the funders played any role in the study design, data collection and analysis, decision to publish, or preparation of the manuscript.

9) Kindly revise your competing statement in the online submission form to align with the journal's style guidelines: 'The authors declare that there are no competing interests.'

**Reviewers' comments:**

Reviewer's Responses to Questions

**Comments to the Authors:**

Reviewer #1: This manuscript presents a comprehensive benchmark study evaluating semi-supervised scRNA-seq integration methods under realistic label imperfections. While the study addresses an important topic and demonstrates rigorous experimental design, several critical limitations prevent me from recommending acceptance in the current form. The manuscript would benefit from additional analyses and methodological refinements to fully address the practical challenges of scRNA-seq integration.

Major Comments：

1. The benchmark should consider one of the most common real-world challenges: inconsistent label schemas across datasets. In practice, researchers frequently encounter datasets with varying annotation granularity or different naming conventions for identical cell types. The current study focuses solely on label imperfections within individual datasets, missing the critical dimension of cross-dataset label harmonization. I strongly recommend adding a new scenario that evaluates methods' performance when integrating datasets with intentionally mismatched label schemas, such as combining datasets with coarse-grained annotations against others with fine-grained subtypes. This expansion would provide crucial insights into how methods handle hierarchical label relationships and whether they can leverage partial label matches, significantly enhancing the practical relevance of your benchmark.

2. The manuscript provides extensive performance comparisons but lacks deep analysis of why certain methods succeed or fail under specific conditions. This limits the utility for method developers seeking to improve future algorithms. Dedicated discussions are needed to analyze the relationship between methodological designs and observed performance patterns.

3. The benchmark identifies limitations but provides minimal actionable guidance for developing improved semi-supervised methods. The conclusion section should be expanded to include specific recommendations for future method design. These concrete suggestions would transform your findings into practical guidance that directly informs the next generation of integration tools.

4. The evaluation of auto-annotation tools is limited to traditional methods, missing important recent advances in deep learning-based approaches. The study should include additional deep learning-based auto-annotation tools in Scenario V. This expansion should compare integration performance when using labels from modern versus traditional auto-annotation methods and analyze whether improved auto-annotation quality translates to better integration outcomes, providing a more comprehensive assessment of current practical workflows.

5. In Figure S7, UMAP plots should be provided where cells are colored by their cell type labels. This visualization will help to intuitively identify the specific cell populations or types where label mismatches most frequently occur.

Minor Comments:

1. In Supplementary Note 7, line 305, the missing references "Figure ??" and "Table ??" need to be corrected to the appropriate figure and table numbers.

Reviewer #2: Semi-supervised integration methods aim to leverage cell type labels to improve batch correction while preserving biological signals. However, current evaluations of their advantages are typically conducted under idealized conditions, whereas real-world datasets often contain imperfect or erroneous labels. The article "A Benchmark of Semi-Supervised scRNA-seq Integration Methods in Real-World Scenarios" provides valuable insights by systematically evaluating six real datasets to assess the performance of semi-supervised single-cell RNA sequencing integration methods compared to unsupervised approaches. The key finding—that semi-supervised methods offer limited benefits when label quality is suboptimal—provides crucial practical guidance. This work establishes a robust evaluation framework for semi-supervised integration methods and offers valuable recommendations for selecting methods in practical data analysis pipelines. However, several aspects remain unclear and require further clarification:

In real-world data, rare cell types may be present in only a few batches, and semi-supervised methods have the potential to preserve signals from these cells. However, this scenario does not seem to be addressed in the paper. Could experiments or metrics be designed to evaluate whether semi-supervised methods offer an advantage in mitigating issues related to excessive batch correction?

Could the computational efficiency of each method be provided to assist in method selection for practical applications?

The text includes references numbered 44, 45, etc., but the reference list only goes up to number 33. It is unclear whether the citations can be matched with the reference list. Perhaps a page is missing from the article. Please verify this issue and also ensure consistency in citation formatting, such as the capitalization of journal names.

The scaling procedure for the scatter plots is not explained very clearly. Since unsupervised methods serve as the baseline, why is their score not equal to 1 in the figure? I would appreciate further clarification on this issue.

Are the metrics calculated in the paper based on the integrated latent space or the integrated expression matrix? For methods that provide an integrated expression matrix, do semi-supervised integration methods help preserve the expression features of important marker genes more effectively? If possible, I would appreciate examples to illustrate this.

Reviewer #3: This paper is a useful benchmark. The authors test scRNA-seq integration methods using realistic imperfect labels, which is a key contribution. The experimental settings are comprehensive, making the conclusions credible. The main finding is important: semi-supervised methods are not very robust and often don't beat the best unsupervised ones in these real-world tests. I recommend Minor Revision.

Major Concern:

The benchmark only checks for accuracy and robustness. They do not compare the speed or memory (computational performance) of the methods. This is an important factor for users.

Minor Concerns:

1. Methods Section (4.2): This section is confusing. They group metrics into two types in 4.2.1, but then list them all together. Please reorganize this. Use two subheadings (one for "bio-conservation" and one for "batch removal") and put the metric definitions under the correct one. This is easier to read.

2. Typos in Supplement: I saw "???" in the supplementary file instead of numbers (Page 12). Please proofread and fix all typos.

**Have the authors made all data and (if applicable) computational code underlying the findings in their manuscript fully available?**

The PLOS Data policy requires authors to make all data and code underlying the findings described in their manuscript fully available without restriction, with rare exception (please refer to the Data Availability Statement in the manuscript PDF file). The data and code should be provided as part of the manuscript or its supporting information, or deposited to a public repository. For example, in addition to summary statistics, the data points behind means, medians and variance measures should be available. If there are restrictions on publicly sharing data or code —e.g. participant privacy or use of data from a third party—those must be specified.requires authors to make all data and code underlying the findings described in their manuscript fully available without restriction, with rare exception (please refer to the Data Availability Statement in the manuscript PDF file). The data and code should be provided as part of the manuscript or its supporting information, or deposited to a public repository. For example, in addition to summary statistics, the data points behind means, medians and variance measures should be available. If there are restrictions on publicly sharing data or code —e.g. participant privacy or use of data from a third party—those must be specified.

Reviewer #1: None

Reviewer #2: None

Reviewer #3: Yes

PLOS authors have the option to publish the peer review history of their article (what does this mean?. If published, this will include your full peer review and any attached files.). If published, this will include your full peer review and any attached files.

**Do you want your identity to be public for this peer review?** For information about this choice, including consent withdrawal, please see our For information about this choice, including consent withdrawal, please see our Privacy Policy ..

Reviewer #1: No

Reviewer #2: No

Reviewer #3: No

**Figure resubmission:**

**Reproducibility:**



---

## [Decision Letter · Decision Letter 1]

10 Feb 2026

Dear Mr. He,

We are pleased to inform you that your manuscript 'A Benchmark of Semi-Supervised scRNA-seq Integration Methods in Real-World Scenarios' has been provisionally accepted for publication in PLOS Computational Biology.

Best regards,

Tao Wang

Academic Editor

PLOS Computational Biology

Ferhat Ay

Section Editor

PLOS Computational Biology

The authors have successfully addressed all reviewers’ comments and revised the manuscript accordingly. I recommend that the manuscript be accepted in its current form.

Reviewer's Responses to Questions

**Comments to the Authors:**

Reviewer #1: The authors have addressed all my comments.

Reviewer #2: The authors have addressed my concerns. I am satisfied with the revised version.

Reviewer #3: Thank you to the authors for their hard work on the revisions, all my concerns have been resolved.

**Have the authors made all data and (if applicable) computational code underlying the findings in their manuscript fully available?**

The PLOS Data policy requires authors to make all data and code underlying the findings described in their manuscript fully available without restriction, with rare exception (please refer to the Data Availability Statement in the manuscript PDF file). The data and code should be provided as part of the manuscript or its supporting information, or deposited to a public repository. For example, in addition to summary statistics, the data points behind means, medians and variance measures should be available. If there are restrictions on publicly sharing data or code —e.g. participant privacy or use of data from a third party—those must be specified.requires authors to make all data and code underlying the findings described in their manuscript fully available without restriction, with rare exception (please refer to the Data Availability Statement in the manuscript PDF file). The data and code should be provided as part of the manuscript or its supporting information, or deposited to a public repository. For example, in addition to summary statistics, the data points behind means, medians and variance measures should be available. If there are restrictions on publicly sharing data or code —e.g. participant privacy or use of data from a third party—those must be specified.

Reviewer #1: None

Reviewer #2: None

Reviewer #3: Yes

PLOS authors have the option to publish the peer review history of their article (what does this mean?. If published, this will include your full peer review and any attached files.). If published, this will include your full peer review and any attached files.

Reviewer #1: No

Reviewer #2: **Yes:** Tao WangTao Wang

Reviewer #3: No

---

## [Editor Report · Acceptance letter]

PCOMPBIOL-D-25-02003R1

A Benchmark of Semi-Supervised scRNA-seq Integration Methods in Real-World Scenarios

Dear Dr He,

I am pleased to inform you that your manuscript has been formally accepted for publication in PLOS Computational Biology. Your manuscript is now with our production department and you will be notified of the publication date in due course.

With kind regards,

Anita Estes
